



# Assessing national exposure and impact to glacial lake outburst floods considering uncertainty under data sparsity

Huili Chen[1], Qiuhua Liang[1], Jiaheng Zhao[2], Sudan Bikash Maharjan[3]

[1] School of Architecture, Building and Civil Engineering, Loughborough University, Loughborough LE11 3TU, UK

[2] FM Global, 117369, Singapore

[3] International Centre for Integrated Mountain Development (ICIMOD), Nepal

*Correspondence to*: Qiuhua Liang (Q.Liang@lboro.ac.uk)

**Abstract.** Glacial Lake Outburst Floods (GLOFs) are widely recognized as one of the most devastating natural hazards in the Himalaya, which may catastrophic consequences including substantial loss of lives. To effectively mitigate these risks and
enhance regional resilience, it is imperative to conduct an objective and holistic assessment of GLOFs hazards and their potential impacts of GLOFs over a large spatial scale. However, this is challenged by the limited availability of data and the inaccessibility to most of the glacial lakes in high-altitude areas. The data challenge is exacerbated when dealing with multiple lakes across an expansive spatial area. This study aims to exploit remote sensing techniques, well-established Bayesian regression models for estimating glacial lake conditions, cutting-edge flood modelling technology, and open data from various
sources to innovate a framework for assessing the national exposure and impact of GLOFs. In the innovative framework, multi-temporal imagery is utilized with a Random Forest model to extract glacial lake water surfaces. Bayesian models, derived from previous research, are employed to estimate a plausible range of glacial lake water volumes and associated GLOF peak discharges, while accounting for the uncertainty stemming from the limited size of available data and outliers within the data. A significant amount of GLOF scenarios is subsequently generated based on this estimated plausible range of peak discharges.
A GPU-based hydrodynamic model is then adopted to simulate the resulting flood hydrodynamics in different GLOF scenarios. Necessary socio-economic information is collected and processed from multiple sources including OpenStreetMap, Google Earth, local archives, and global data products to support exposure analysis. Established depth-damage curves are used to assess the GLOF damage extents to different exposures. The evaluation framework is applied to 21 glacial lakes identified potentially dangerous in the Nepal Himalaya. The results indicate that Tsho Rolpa Lake, Lower Barun Lake and Thulagi Lake
bear the most serious impacts of GLOFs on buildings and roads, and influence existing hydropower facilities, while Lower Barun Lake, Tsho Rolpa Lake and Lumding Lake will experience the most impacts of GLOFs on agriculture areas. Four anonymous lakes (located at 85°37′51″ E, 28°09′44″ N; 87°44′59″ E, 27°48′57″ N; 87°56′05″ N, 27°47′26″ E; 86°55′41″ E, 27°51′00″ N) have the potential to impact more than 100 buildings, and the first three lakes may even submerge existing hydropower facilities.

## 1 Introduction

Glacial Lake Outburst Floods (GLOFs) are recognized as one of the most impactful natural hazards in the Himalaya, where these disasters have had the highest death toll worldwide and caused serious economic damages (Veh et al., 2020). GLOFs can generate transient discharges that are orders of magnitude greater than the typical annual floods in the receiving rivers (Cenderelli and Wohl, 2001) and some of them can travel >200 km downstream (Richardson & Reynolds, 2000). The extreme
discharges, accelerating along the steep mountainous terrains, make GLOFs extremely destructive to downstream communities and infrastructure systems. The unpredictable nature of GLOFs, often occurring without warning, has left downstream communities and infrastructure ill-prepared, causing the loss of human lives and economic damages. The ongoing impact of climate change has introduced additional uncertainty into GLOF risk. The Himalaya region has observed extensive glacier



shrinkage and a proliferation of glacial lakes (Zhang et al., 2015). The potential impacts of GLOFs on downstream communities

are expected to intensify further due to population growth and socio-economic development. Hence, it is crucial to develop effective strategies for managing GLOF risks to enhance human safety and support sustainable development. This necessitates an objective and reproducible assessment of GLOF hazards and their potential impacts arising from these glacial lakes.

Some potentially dangerous lakes have been well-studied individually, such as Tsho Rolpa Lake (e.g., Shrestha & Nakagawa, 2014), Imja Tsho lake (e.g., Somos-Valenzuela et al., 2015), and Lower Barun Lake (e.g., Sattar et al., 2021). However, these

studies typically focus on individual glacial lakes, which provide limited insight into the overall danger and potential impacts of glacial lakes as a collective whole. While there have been assessments of glacial lake hazards in the Himalayan region, certain limitations exist. Previous work by Mool et al., (2011) and Bajracharya et al., (2020) employed remote sensing techniques to identify potentially dangerous glacial lakes (PDGLs) in Nepal, considering different hazard factors. Rounce et al. (2017) undertook a similar study, quantifying the hazard level of 131 glacial lakes with > 0.1 km$^2$ area in Nepal.

Furthermore, Rounce et al. (2017) evaluated the potential downstream impacts of GLOFs caused by these glacial lakes using a simple flood model without any physical basis. This simple flood model has been also applied to evaluate the overall impacts of GLOFs originating from multiple glacial lakes in the Indian Himalaya (Dubey & Goyal, 2020). Zheng et al. (2021) extended their analysis to assess the impacts of GLOFs across the Third Pole by using a Geographic Information System (GIS)-based modified single-flow hydrological model. However, the complexity of GLOFs, characterized by complex hydraulic dynamics

resulting from sudden releases of large water volumes and the rugged, steep terrain downstream, renders simple flood models insufficient for capturing the complex dynamics of GLOFs to support a detailed assessment of the potential impacts on downstream communities and their infrastructure.

A range of physically based hydrodynamic models has been developed and applied to predict the spatial-temporal process of GLOFs, offering detailed insights into the resulting flood impacts (e.g., Worni et al., 2014; Ancey et al., 2019; Sattar et al.,

2019). However, they entail a huge amount of computation and face substantial demands for computation resource when applied at a large scale. What's even more challenging is that the computational requirements increase significantly when addressing GLOF simulations involving multiple scenarios for multiple glacial lakes. Moreover, the application of hydrodynamic models to support GLOF modelling and impact assessment necessitates a considerable amount of data.

Data availability poses a significant challenge. The high-alpine conditions have constrained our ability to acquire detailed

spatial data for multiple lakes across a large scale. To correctly depict dynamic inundation process of GLOFs, glacial lake conditions and dam breaching process are essential to estimate the outflow discharge resulting from a breach. The outflow discharge hydrograph serves as input for hydrodynamic models, enabling predictions of spatiotemporal changes in flood dynamics. While the distribution and changes of glacial lakes have been extensively mapped in recent years (e.g., Zhang et al., 2015; Nie et al., 2017; Shugar et al., 2020), accurately determining lake volume and reliably predicting dam breaching

processes has remained a challenge. The high-alpine conditions impede detailed fieldwork, leading researchers to delineate glacial lakes from increasingly detailed digital topographic data and satellite imagery. Combining these available data with existing lake bathymetry measurements offers the possibility of estimating water volumes and peak discharges from outbursts by establishing empirical relationships. However, estimated lake volumes and potential peak discharges derived from these empirical relationships can vary by up to an order of magnitude (Cook and Quincey, 2015; Muñoz et al., 2020). To account

for the uncertainties inherent in conventional empirical relationships, Veh et al. (2020) developed a Bayesian robust regression, utilizing data from the bathymetric survey of 24 glacial lakes. This model estimates water volume based on the surface areas of glacial lakes. Simultaneously, they created a Bayesian variant of a physical dam-break model originally proposed by Walder & O'Connor (1997) to predict peak discharge associated with the estimated flood volume. The Bayesian estimates explore the parameter space of plausible flood volumes and associated peak discharges, generating approximately a million outburst

scenarios for each lake. These scenarios comprehensively consider all potential conditions of the dam breach process for each





specific lake and provide a full range of input information for hydrodynamic models, thereby facilitating predictions of the GLOF inundation process. Therefore, this study aims to leverage these established Bayesian models to generate a comprehensive set of outburst scenarios for the glacial lakes of interest. The independent variable in these Bayesian models is the glacial lake area, which can be estimated using remote sensing techniques.

GLOF exposure and impact assessment is also restricted by data sparsity. Previous studies have typically relied on census data at coarse spatial resolutions or aggregated land use data that encompass various objects like properties and infrastructure, to estimate the potential socio-economic impact of GLOFs. Benefiting from the emergence of new data technologies and the resulting enhancements in data quantity and quality, a spatially explicit assessment method has been developed to identify GLOF exposure at an object level and applied to the Tsho Rolpa Lake (Chen et al., 2022). Employing a similar strategy, 90 essential socio-economic information is collected and processed from various sources, including OpenStreetMap, Google Earth, global data products, and local archives. The information is used to create a spatial exposure dataset that specifies the locations of different objects, such as individual buildings and hydropower facilities. Subsequently, this spatial exposure data is overlaid with the spatially distributed flood simulation outputs to identify potential exposure to GLOFs along their path.

Overall, this study aims to explore the use of remote sensing techniques, the developed Bayesian regression models for 95 estimating lake volumes and potential peak discharges, a physically based hydrodynamic model supported by parallelized high-performance computing, and socio-economic information from multiple sources, to facilitate object-based exposure and potential impact assessments of GLOFs for multiple lakes across a large scale. Nepal has been chosen as the test area due to its abundance of glacial lakes, and it has been reported to experience the most significant national-level economic consequences from GLOFs globally (Carrivick & Tweed, 2016). Section 2 will present the GLOF exposure and impact assessment 100 framework for glacial lakes at a national scale. Section 3 introduces the case study, while Section 4 presents the results. Further discussion will be found in Section 5, and Section 6 will offer concise conclusions drawn from the study.

## 2 Methodology and data

The proposed framework for object-based exposure and impact assessment of GLOFs across multiple lakes comprises several key components: extraction of glacial lake water surfaces from multi-temporal imagery, estimation of lake volumes and peak 105 discharges using well-established Bayesian regression models, utilization of a high-performance hydrodynamic flood model accelerated by GPU technology, and the creation of an exposure dataset sourced from open-source data (Fig. 1). In particular, leveraging multi-temporal imagery availability, a Random Forest model is developed using a set of predictor variables to delineate the maximum extent of glacial lake water surfaces. The plausible range of glacial lake water depths, volumes, and GLOF-induced peak discharges is estimated through existing Bayesian models. A substantial number of GLOF scenarios, 110 encompassing outflow discharge hydrographs through glacial lakes, are sampled and generated based on the plausible range of peak discharges. For each scenario, the resulting outflow discharge hydrograph is employed to drive the GPU-accelerated hydrodynamic model, efficiently simulating the temporal and spatial dynamics of floods. These flood dynamics are then overlaid with the spatial exposure data to identify potential exposure to GLOFs and to quantify damage extent by using established depth-damage curves.



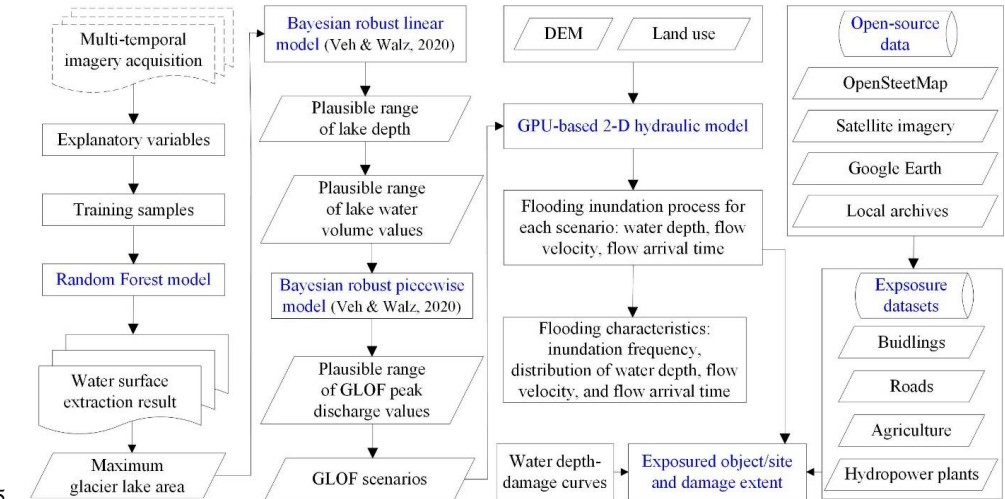

**Fig. 1. GLOF exposure and impact assessment framework for multiple glacial lakes (key components highlighted in blue)**

## 2.1 Glacial lake water surface extraction

With the availability of multi-temporal imagery, a Random Forest model based on a set of predictor variables is used to map the location and extent of water surfaces of glacial lakes under different hydrological conditions in order to produce the maximum extent of lake water surfaces.

### 2.1.1 Acquisition of satellite imagery

Sentinel-2 is operational multispectral imaging mission of the European Space Agency for global land observation. The Sentinel-2A and -2B satellites were launched in 2015 and 2017, respectively. These satellites capture imagery every 10 days (every 5 days with the twin satellites together). The spatial resolution for the visible and broad Near Infrared (NIR) bands is 10 meters, while it is 20 meters for the red edge, narrow NIR, and Short-wave Infrared (SWIR) bands. Here, all available Sentinel-2 imagery for the case study glacial lakes is utilized to identify the maximum extent of their water surfaces.

The analysis is based on the Sentinel-2 level-1C Top-Of-Atmosphere (TOA) products, which are accessible through the Google Earth Engine. Any observations affected by clouds are masked using the Sentinel-2 Quality Assurance (QA) band flags. Bands originally at a 20-meter resolution are resampled to 10 m using the nearest neighbour method before being stacked for subsequent interpretation. All available Sentinel-2 datasets are collected and filtered to reserve imagery from the ablation season, reducing the impact of frozen water surfaces, as per the empirical period of the local melt season (Shugar et al., 2020). In total, 1,520 Sentinel-2 images have been collected for this purpose.

### 2.1.2 Random Forest model

Mapping water surfaces from multiple images is a complex task that necessitates the consideration and analysis of various water-related signals in spectral responses, often influenced by water turbidity and bottom sediments. In this context, a Random Forest model is developed based on a set of predictor variables to extract water surfaces. Random Forest modelling is an ensemble classification technique (Breiman, 2001) and has been extensively used in the classification of remote sensing data (e.g., Yu et al., 2011; Rodriguez-Galiano et al., 2012). Random Forest models excel at recognizing regional variations in threshold values, surpassing the capabilities of traditional index thresholding methods (Tulbure et al., 2016). Notably, Random Forest models do not rely on data distribution assumptions and can yield accurate predictions without overfitting data.





Consequently, they have been increasingly used in water surface extraction as a favourable alternative to the traditional statistical approaches (e.g., Schaffer-Smith et al., 2017; Veh et al., 2018).

Random Forest model consists of a set of classification trees, each of which grows from a random subset of training samples and randomly permuted explanatory variables. The classification trees can grow to a specified maximum number without pruning, and the final classifications are determined by the majority votes of the trees in the forest. The explanatory variables for Sentinel-2 datasets in the Random Forest model include TOA reflectance for every spectral band, brightness temperature, vegetation indices, and water indices. TOA reflectance and brightness temperature are obtained by normalizing the target imagery, mitigating unwanted effects resulting from variations in sun angle and earth-sun distance. The vegetation indices include the Normalized Difference Vegetation Index (NDVI) and the Enhanced Vegetation Index (EVI). NDVI is sensitive to chlorophyll and used to assess terrestrial vegetation conditions (Tucker, 1979) while EVI is developed to optimize the vegetation signal in high biomass regions, de-couple canopy background signal, and reduce atmosphere influences (Huete et al., 2002). Water indices include the Normalized Difference Water Index (NDWI, McFeeters, 1996), Modified NDWI (MNDWI, Xu, 2006), and Normalized Difference Moisture Index (NDMI, Gao, 1996). NDWI enhances the response to open water features while minimizing soil and terrestrial vegetation influences. MNDWI substitutes the middle infrared band for the near infrared band used in the NDWI to enhance water features and remove the noises from other land types. NDMI is an effective indicator of vegetation water content. The training samples are selected via visual interpretation of satellite images to represent glacial lake water surfaces, along with various non-water covers, including diverse landscapes and vegetation types. The uncertainty in estimating glacial lake area is quantified using a widely used buffer method (Granshaw and Fountain, 2006). A buffer area of half a pixel (e.g., Zhang et al., 2015; Krause et al., 2019) is adopted to measure the uncertainty in estimated lake area. The misclassified glacial lake water areas resulting from terrain shadows are eliminated during post-processing, through manual exclusion of inaccurately classified regions.

## 2.2 GLOF dynamic inundation process simulation

Using the maximum extent of glacial lake water surfaces, we employ the established Bayesian models to predict glacial lake conditions and the dam breaching process. This allows us to estimate the full range of GLOF outflow discharge through the breach. Subsequently, various GLOF scenarios featuring a range of outflow discharge hydrographs are then sampled to drive the GPU-based hydrodynamic model for the simulation of dynamic flood dynamics resulting from GLOFs.

### 2.2.1 Estimating volumes and peak discharge of glacial lakes

Global samples from glacial lakes have suggested that the water depths for glacial lakes with similar surface area can vary by one order of magnitude. To estimate water volumes of glacial lakes, we adopted the model that relates lake areas to their maximum depths, which was developed by Veh & Walz (2020). The model was built by compiling the reported lake areas and maximum depths obtained from bathymetric surveys conducted on 24 Himalayan glacial lakes. A Bayesian robust linear regression with a normally distributed target variable (lake depth $d$) $d \sim N(\mu_d(a), 1/\tau)$ is adopted to account for possible effects of the limited sample size and outliers present in the compiled dataset. The mean $\mu_d(a)$ is caculated below through a linear combination of the input lake area a. The precision $\tau$ (the inverse of variance) is gamma-distributed $\tau \sim \Gamma(0.001,\ 0.001)$.

$$\mu_d(a) = \alpha_0 + \alpha_1 a \tag{1}$$

Where $a$ is lake area, intercept $\alpha_0 \sim N(0,\ 10^{-12})$, slope $\alpha_1 \sim N(0,\ 10^{-12})$.

We obtained 100 posterior estimates for the lake depth ($d$) from the Bayesian model for each lake. For each lake, samples inside the 95 % highest density interval (HDI) of credible lake depth values are reserved, i.e., 94 lake depth samples for each





lake. In this study, we maintained the same assumption regarding the bathymetry of the glacial lakes as outlined by Veh & Walz (2020). The delineated lake from satellite imagery is circular and each lake is assumed an ellipsoidal bathymetry. Therefore, we obtained 94 estimates of total volume ($V_{tot}$) for each glacial lake.

$$V_{tot} = (2/3)\,da \tag{2}$$

With regard to estimating peak discharge during dam failure, Veh & Walz (2020) built a Bayesian piecewise robust model to

characterize the physically motivated model developed by Walder & O'Connor (1997). The latter model predicts peak discharge $Q_p$ during natural dam failure. In their study, Walder & O'Connor (1997) compiled data from 63 observed natural dam breaks in various settings and identified a constant response of dimensionless peak discharge $Qp*$ when plotted against the dimensionless product $\eta$ of lake volume and breach rate $k$. They inferred a model that describes the relationship between peak discharge and lake volume using the dimensionless peak discharge $Qp*$.

$$Q_p{}^* = Q_P g^{-\frac{1}{2}} h^{-\frac{5}{2}} \tag{3}$$

$$\eta = V_O^* k^* \tag{4}$$

Where $V_O^* = V_0 h^{-3}$ represents the dimensionless flood volume, $k^* = k g^{-1/2} h^{-1/2}$ is the dimensionless breach rate, $g$ is the acceleration of gravity, $h$ is breach depth, and $V_0$ is the released water volume. $k$ is the breach rate and subsumes lithologic conditions, the erodibility of the outflow channel, and the breach and downstream valley geometry. $h$ is measured from the

final lake surface after dam failure to the initial lake surface. $V_0$ is the released water volume and depends on $h$ and $V_{tot}$.

Empirical data support a piecewise regression model in the form $Q_p{}^* = b_0 \eta^{b_1}$ ($b_0$ and $b_1$ are the regression parameters) for $\eta < \eta_c$, and $Q_p{}^*$ is constant for $\eta > \eta_c$. Bayesian piecewise linear regression was developed for predicting peak discharge $Q_p{}^*$ from $\eta$, the product of breach rate $k$ and released flood volume (Veh & Walz, 2020).

### 2.2.2 2-D hydrodynamic modelling

A fully dynamic model based on the 2-D depth-averaged shallow water equations (SWEs) is adopted to route the breach hydrograph. The conservative form of the governing 2-D shallow water equations is expressed as follows:

$$\frac{\partial \mathbf{q}}{\partial t} + \frac{\partial \mathbf{f}}{\partial x} + \frac{\partial \mathbf{g}}{\partial y} = \mathbf{s} \tag{5}$$

where $t$ is the time; $x$ and $y$ represent the Cartesian coordinates; $\mathbf{q}$ denotes the flow variable vector; $\mathbf{f}$ and $\mathbf{g}$ are the flux vectors in the $x$- and $y$-direction, respectively; and $\mathbf{s}$ is the source term vector. The vector terms are defined as:

$$\mathbf{q} = \begin{bmatrix} h \\ q_x \\ q_y \end{bmatrix} \qquad \mathbf{f} = \begin{bmatrix} q_x \\ uq_x + \frac{1}{2}gh^2 \\ uq_y \end{bmatrix}$$

$$\mathbf{g} = \begin{bmatrix} q_y \\ vq_x \\ vq_y + \frac{1}{2}gh^2 \end{bmatrix} \qquad \mathbf{s} = \begin{bmatrix} 0 \\ -C_f u\sqrt{u^2 + v^2} - gh\frac{\partial z_b}{\partial x} \\ -C_f v\sqrt{u^2 + v^2} - gh\frac{\partial z_b}{\partial y} \end{bmatrix} \tag{6}$$

where $h$ is the water depth; $q_x = uh$ and $q_y = vh$ are the unit-width discharges in the $x$- and $y$- directions, respectively; $u$ and $v$ denote the depth-averaged velocities in two Cartesian directions; and $z_b$ is the bed elevation; and $C_f$ is the bed roughness coefficient.

The governing equations outlined above are solved through a shock-capturing finite volume Godunov-type scheme on uniform grids (Zhao & Liang, 2022). The numerical scheme introduces a robust Godunov-type model to deliver precise and stable



predictions of overland flow and flooding processes at the catchment scale. This novel scheme is employed to enhance the High-Performance Integrated Hydrodynamic Modelling System (HiPIMS) and implemented through a Python and CUDA C hybrid programming framework to achieve multi-GPU and multi-node high-performance computing for large-scale

simulations. It's worth noting that the GPU-accelerated model has demonstrated computational efficiency up to ten times greater than its CPU-based counterpart. The 2-D hydrodynamic model is set up using the terrain data and roughness data, and it is driven by breach hydrograph for each scenario, as calculated in Section 2.2.1. Subsequently, the runoff is automatically routed throughout the flow area.

### 2.3 GLOF exposure and impact assessment

Based on the GLOF inundation process predicted by HiPIMS for each scenario, which includes water depth, flow velocities, and flood arrival time, we can estimate potential flood exposure by superimposing the exposure datasets onto the flood simulation results. In addition to assessing flood exposure, it is imperative to quantify the potential losses and impacts of GLOFs under various conditions to understand the associated risks. Estimating the direct damage to buildings and other exposed objects can be achieved by employing appropriate depth-damage curves that establish the relationship between flood

depth and the potential damage. Typically, the damage is quantified as a percentage of the cost required for repairs or replacements. In this study, we utilize depth-damage curves from the HAZUS Flood model to investigate the impact of GLOFs on buildings (Scawthorn et al., 2006). Beyond buildings, GLOFs can also have significant impact on agricultural lands and roads. We evaluate the damage to agricultural lands and roads caused by GLOFs using the damage curves recommended in a technical report published by the Joint Research Centre of the European Commission (Huizinga et al., 2017). The specific

water depth-damage curves for buildings, roads, and agricultural lands used in this study can be referenced in Chen et al. (2023).

### 2.4 Data

HiPIMS is set up using a digital elevation model (DEM) to represent domain topography and land use data to parameterise domain roughness. It is driven by the out-of-breach flow discharge estimated in Section 2.2.1. The DEM used in this work is

Shuttle Radar Topography Mission (SRTM) DEM with a spatial resolution of 30 m (Farr et al., 2007). Land use types are extracted from the Landsat TM imagery from the year 2010, provided by the International Centre for Integrated Mountain Development (ICIMOD, 2020). Roughness of the flow area is represented by the Manning coefficient ($n$), which is dependent on land use types. The Manning coefficients are specified according to previous hydraulics textbooks or reports (e.g., Chow, 1959, Barnes, 1967, Arcement and Schneider, 1984). The values assigned are 0.15 for forest, 0.035 for arable land, 0.03 for

grassland, 0.027 for water surface, and 0.016 for construction land.

Open-source datasets are used to support the assessment of GLOF exposure and impacts. The OpenStreetMap (OSM) is a collaborative user-generated project initiated in 2004 to provide an openly available geographical database of the world, covering both the natural and artificial environment of the Earth's surface (OpenStreetMap contributors, 2015). While primarily built by volunteers, OSM also integrates geographical data contributed by governmental and specialized GIS databases for

certain areas or entire countries, e.g., Nepal, providing relatively complete spatial data on buildings and other objects. Hydropower plant data are obtained from Hydro Map project (Nepal Hydropower Portal, 2019). In Hydro Map project, hydropower plants are categorized into three types: Operation, Generation and Survey. In Nepal, the hydropower licensing regime is divided into two stages i.e., a survey license is issued to conduct a feasibility and environmental assessment, and a generation license is granted after the project is found to be technically, environment friendly, and economically viable.

Detailed information on each hydropower plant is provided, including its Province, District, Local Government, capacity, commission/issue date, longitude, and latitude etc. Importing hydropower plant data in ArcGIS and comparing with sub-meter imagery from ArcGIS Server and Google Earth, the positions of some hydropower plants are found to be inaccurate. To address





the inaccuracies in the positions of some hydropower plants, a process has been undertaken to enhance the quality of the hydropower plant data. The coordinates of existing hydropower plants, including those in operation and under construction,

have been collected from Wikipedia. These coordinates are then visually inspected and collected against sub-meter imagery obtained from ArcGIS Server and Google Earth, as they are discernible in sub-meter imagery. The newly collected coordinates will be utilized to update the spatial positions of hydropower plants provided by the Hydro Map project.

## 3 Study area and glacial lakes

Nepal is highly vulnerable to GLOFs. A total of 53 GLOF events have been documented in Nepal from 1560 to now (Shrestha

et al., 2023). Additionally, there have been 37 GLOF events recorded in the Tibetan Autonomous Region, China, which had transboundary impacts on Nepal. These historical events have brought devastating consequences to the country. For example, both the 1985 Dig Tsho GLOF and the 1998 Tam pokhari GLOF had devastating effects, resulting in significant loss of life, property, infrastructure damage, and severe disruptions to the livelihoods of those living in downstream areas. Approximately 1.56 million people live downstream within 3 km of moraine-dammed lakes in Nepal, putting them at risk of GLOFs (Ghimire,

2004). If climate change continues at its present pace, rates of glacier mass loss and shrinkage, and the formation and expansion of glacial lakes will increase further, which could escalate the occurrence of GLOFs. Exacerbating the situation, GLOF exposure and risk are on the rise due to the expansion of settlements, economic activities, and infrastructure construction along riverbanks.

In Nepal, a total of 2,070 glacial lakes with lake area equal to or larger than 0.003 km$^2$ have been identified and mapped by

using Landsat images (Bajracharya et al, 2020). These glacial lakes are predominantly situated in northern Nepal, at elevations ranging from 3400m to 5908m. Notably, 98% of these glacial lakes are positioned above 4000m. Bajracharya et al. (2020) assessed GLOF hazard factors related to lake and dam characteristics, glacier activity at the source, and the morphology of the lake surroundings for the 2,070 glacial lakes. They identified 21 lakes as potentially dangerous glacial lakes (PDGLs) (Fig 2 and Table 1). Among the 21 PDGLs, some lakes have names, while others do not and were designated as 'Anonymous *'.

These 21 PDGLs are further categorized into three ranks based on the level of danger associated with GLOF hazards, with Rank I representing the highest level of risk. Among the 21 PDGLs, 15 lakes were classified as Rank I, 3 as Rank II, and 3 lakes as Rank III. Lakes are classified as Rank I due to high-risk factors, such as large lake size, the potential for expansion caused by glacier calving, steep outlet slopes, and the likelihood of snow and/or ice avalanches and landslides in the surrounding areas. For example, Tsho Rolpa Lake, classified as Rank I, is a typical example of moraine-dammed lakes that

have formed from supraglacial lakes. It is bordered by steep-slope lateral moraine (25-80°) and dammed by a partially ice-cored end moraine (8.5-16.7°). The end moraine is composed of unconsolidated sediments comprising boulders, gravel, sand, and silt. Over time, its area and volume have exhibited an increasing trend, while the lake bed has progressively deepened with the retreat of the calving source glacier. These factors collectively classify Tsho Rolpa Lake as one of the Rank I PDGLs in Nepal.





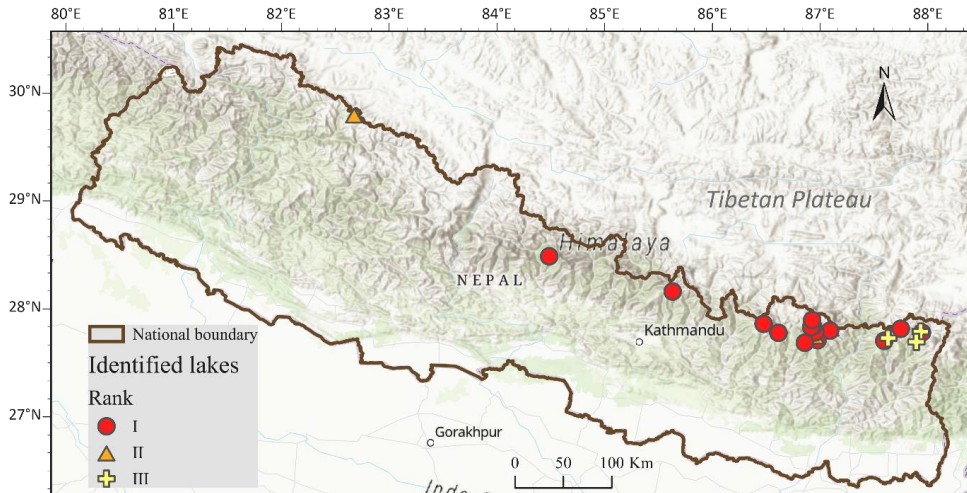


**Fig 2. Study area, and 21 identified dangerous glacial lakes and their danger level rank of GLOF hazards with Rank I being the highest.**

In this study, the focus is on these 21 PDGLs, and a comprehensive assessment of their GLOF risk and downstream impacts is conducted. Each lake is assessed by using the proposed evaluation framework in section 2. The model and evaluation domain

for each lake are determined based on the maximum potential inundation extent resulting from GLOFs, as well as the topographic features and river network conditions downstream. Typically, the domain spans more than 100 km and is sufficiently extensive to encompass all potential impacts.





**Table 1 Delineated glacial lake areas under varied water-occurrence frequency from multi-temporal Sentinel-2 imagery**

| Lake number | Lake ID | Lake name | Longitude (E) | Latitude (N) | Rank | Area (km²) (> 5%) | Area (km²) (> 25%) | Area (km²) (> 50%) |
|---|---|---|---|---|---|---|---|---|
| 1 | GL087749E27816N | Anonymous 1 | 87°44′59″ | 27°48′57″ | I | 0.178 ± 0.011 | 0.169 ± 0.011 | 0.161 ± 0.011 |
| 2 | GL087934E27790N | Anonymous 2 | 87°56′05″ | 27°47′26″ | III | 0.148 ± 0.012 | 0.134 ± 0.012 | 0.112 ± 0.010 |
| 3 | GL087945E27781N | Anonymous 3 | 87°55′42″ | 27°46′51″ | I | 0.048 ± 0.005 | 0.040 ± 0.005 | 0.035 ± 0.004 |
| 4 | GL087632E27729N | Anonymous 4 | 87°37′55″ | 27°43′44″ | III | 0.036 ± 0.004 | 0.032 ± 0.004 | 0.016 ± 0.005 |
| 5 | GL087596E27705N | Anonymous 5 | 87°35′46″ | 27°42′18″ | I | 0.026 ± 0.003 | 0.020 ± 0.003 | 0.010 ± 0.003 |
| 6 | GL087893E27694N | Anonymous 6 | 87°53′36″ | 27°41′41″ | III | 0.037 ± 0.005 | 0.028 ± 0.005 | 0.015 ± 0.004 |
| 7 | GL086925E27898N | Imja Tsho | 86°55′30″ | 27°53′53″ | I | 1.741 ± 0.047 | 1.630 ± 0.042 | 1.561 ± 0.041 |
| 8 | GL086476E27861N | Tsho Rolpa | 86°28′34″ | 27°51′40″ | I | 1.712 ± 0.043 | 1.657 ± 0.041 | 1.610 ± 0.040 |
| 9 | GL086928E27850N | Anonymous 7 | 86°55′41″ | 27°51′00″ | I | 0.553 ± 0.021 | 0.533 ± 0.021 | 0.510 ± 0.022 |
| 10 | GL086935E27838N | Hongu 1 | 86°56′06″ | 27°50′17″ | I | 0.322 ± 0.018 | 0.305 ± 0.018 | 0.293 ± 0.018 |
| 11 | GL086917E27832N | Anonymous 8 | 86°55′01″ | 27°49′55″ | I | 0.361 ± 0.015 | 0.342 ± 0.014 | 0.332 ± 0.014 |
| 12 | GL087095E27829N | Anonymous 9 | 87°05′42″ | 27°49′44″ | II | 0.118 ± 0.008 | 0.114 ± 0.008 | 0.037 ± 0.012 |
| 13 | GL087092E27798N | Lower Barun | 87°05′31″ | 27°47′53″ | I | 2.193 ± 0.048 | 2.044 ± 0.046 | 1.900 ± 0.053 |
| 14 | GL086957E27783N | Hongu 2 | 87°57′25″ | 27°46′59″ | I | 0.872 ± 0.030 | 0.865 ± 0.030 | 0.843 ± 0.030 |
| 15 | GL086612E27779N | Lumding | 86°36′43″ | 27°46′44″ | I | 1.475 ± 0.037 | 1.411 ± 0.034 | 1.349 ± 0.035 |
| 16 | GL086958E27755N | Chamlang | 86°57′29″ | 27°45′18″ | II | 0.921 ± 0.027 | 0.856 ± 0.021 | 0.700 ± 0.026 |
| 17 | GL086977E27711N | Anonymous 10 | 86°58′37″ | 27°42′40″ | I | 0.085 ± 0.007 | 0.074 ± 0.007 | 0.009 ± 0.003 |
| 18 | GL086858E27687N | Anonymous 11 | 86°51′29″ | 27°41′13″ | I | 0.336 ± 0.015 | 0.324 ± 0.015 | 0.307 ± 0.014 |
| 19 | GL085630E28162N | Anonymous 12 | 85°37′51″ | 28°09′44″ | I | 0.150 ± 0.009 | 0.137 ± 0.008 | 0.124 ± 0.008 |
| 20 | GL082673E29802N | Anonymous 13 | 82°40′27″ | 29°48′09″ | II | 0.047 ± 0.006 | 0.041 ± 0.005 | 0.032 ± 0.005 |
| 21 | GL084485E28488N | Thulagi | 84°29′06″ | 28°29′17″ | I | 0.997 ± 0.032 | 0.964 ± 0.032 | 0.921 ± 0.029 |


## 4 Results

### 4.1 Glacial Lake Water Surface Extraction

Water surfaces of glacial lakes are delineated from Sentinel-2 images using the Random Forest model, as previously outlined. The Random Forest model is trained with a set of training samples that comprise both water and non-water features. To account for seasonal variations in lake water surfaces, the training samples for water features are manually selected from images acquired at different times. Various non-water features encompass diverse landscapes and vegetation types. The data collected from each training pixel includes TOA reflectance values for individual spectral bands and various band combinations. This training dataset is subsequently employed to drive and train the Random Forest model. The classification is executed on Google Earth Engine, providing not only access to remote sensing data on a global scale but also harnessing the substantial computing power of Google's cloud infrastructure. The analysis involves the computation of water-occurrence frequency based on multi-temporal water surfaces. The outcomes of water-occurrence frequency for specific representative lakes are visually presented in Fig. 3. It is noteworthy that lake areas are not consistently characterized by open water throughout the year. For instance, lake 'Anonymous 1' (87°44′59″ E, 27°48′57″ N) (Fig. 3(b)) exhibit an average water-occurrence frequency of 72%, while 'Anonymous 2' (87°56′05″ E, 27°47′26″ N) (Fig. 3(d)) has an average water-occurrence frequency of 58%. In contrast, for certain lakes, like 'Anonymous 8' (86°55′01″ E, 27°49′55″ N) and the Tsho Ropla Lake, lake areas are always covered with water. Hence, the capacity to map glacial lakes to assess the associated GLOF risk is influenced by the timing of image acquisition.

Table 1 presents the determined lake areas based on varying water-occurrence frequencies. To mitigate the effects of misinterpretations, such as cloud shadows, a 5% threshold is utilized to exclude areas characterized by low water-occurrence frequencies. Subsequently, the maximum lake boundary is delineated for each lake, allowing for the straightforward calculation of maximum lake areas in ArcGIS. Among the 21 lakes, the largest one is Lower Barun Lake (Fig 3 (a)), a substantial glacial lake in Nepal known for its depth and size. Its area measures $2.193 \pm 0.048$ km$^2$, while the smallest lake (Anonymous 5; 87°35′46″ E, 27°42′18″ N) covers only $0.026 \pm 0.003$ km$^2$. Remarkably, Lower Barun Lake has undergone significant area growth since its initial appearance, with an area of 0.04 km$^2$ in 1987 (Sattar et al., 2021), 0.64 km$^2$ in 1989 (Maskey et al., 2020), 1.79 km$^2$ in 2017 (Haritashya et al., 2018), 2 km$^2$ in 2018 (Maskey et al., 2020), and 2.09 km$^2$ in 2019 (Sattar et al., 2021). Imja Tsho Lake, the second largest PDGL, also underwent rapid growth in both area and volume. It did not exist in 1960, but its area in 1963, 1992, 2002, and 2012 measured 0.03, 0.648, 0.868, and 1.257 km$^2$, respectively (Budhathoki et al., 2010; Somos-Valenzuela et al., 2014). The estimated maximum area of Imja Tsho Lake here is $1.741 \pm 0.047$ km$^2$. Tsho Rolpa Lake boasts a maximum area estimated at $1.712 \pm 0.043$ km$^2$. This aligns with previous findings, which reported that the lake had an area of 0.23 km$^2$ in 1957, which grew to 1.02 km$^2$ in 1979, 1.65 km$^2$ in 1999, and 1.61 km$^2$ in 2019 (Chen et al., 2021). Lumding Lake, another PDGL with an estimated area exceeding 1 km$^2$, displayed notable growth. It had an area of 0.104 km$^2$ in 1963, 0.66 km$^2$ in 1987, 0.8 km$^2$ in 1996, and 1.18 km$^2$ in 2016 (Khadka et al., 2019). Our assessment indicates that the maximum area of Lumding Lake is $1.475 \pm 0.037$ km$^2$. In summary, the estimated maximum lake areas derived from multi-temporal satellite images for these extensively studied lakes are in good agreement with previous research. To establish the maximum lake boundary for potential risk assessment, it is imperative to leverage multi-temporal imagery capturing various hydrological conditions of glacial lakes.

The maximum areas of the four large lakes (Lower Barun, Imja Tsho, Tsho Rolpa, and Lumding), each exceeding 1 km$^2$, are approximately 1.1 times the extents where water covers more than 50% of the time. In contrast, for the comparatively smaller lakes (Anonymous 3, 4, 5, 6, 10, and 13), the ratio of maximum area to the area covered by water for more than 50% of the time can be as high as 1.4 to 2.5 times. For instance, 'Anonymous 10' (86°58′37″ E, 27°42′40″ N) has a maximum area of 0.085 km$^2$, while only 0.009 km$^2$ is covered with water for more than 50% of the time. The areas of small PDGLs exhibit more





significant variations in space and time compared to those of larger PDGLs, making the associated risks more uncertain. Additionally, the ratio of maximum area to the area covered by water for more than 50% of the time is predominantly in the range of 1.1 to 1.5 for PDGLs with high hazard level I. However, for PDGLs with lower hazard levels II and III, this ratio varies from 1.3 to 3.2. This indicates that the areas of PDGLs with a high hazard level exhibit more stability in terms of space

and time compared to those with lower hazard levels.

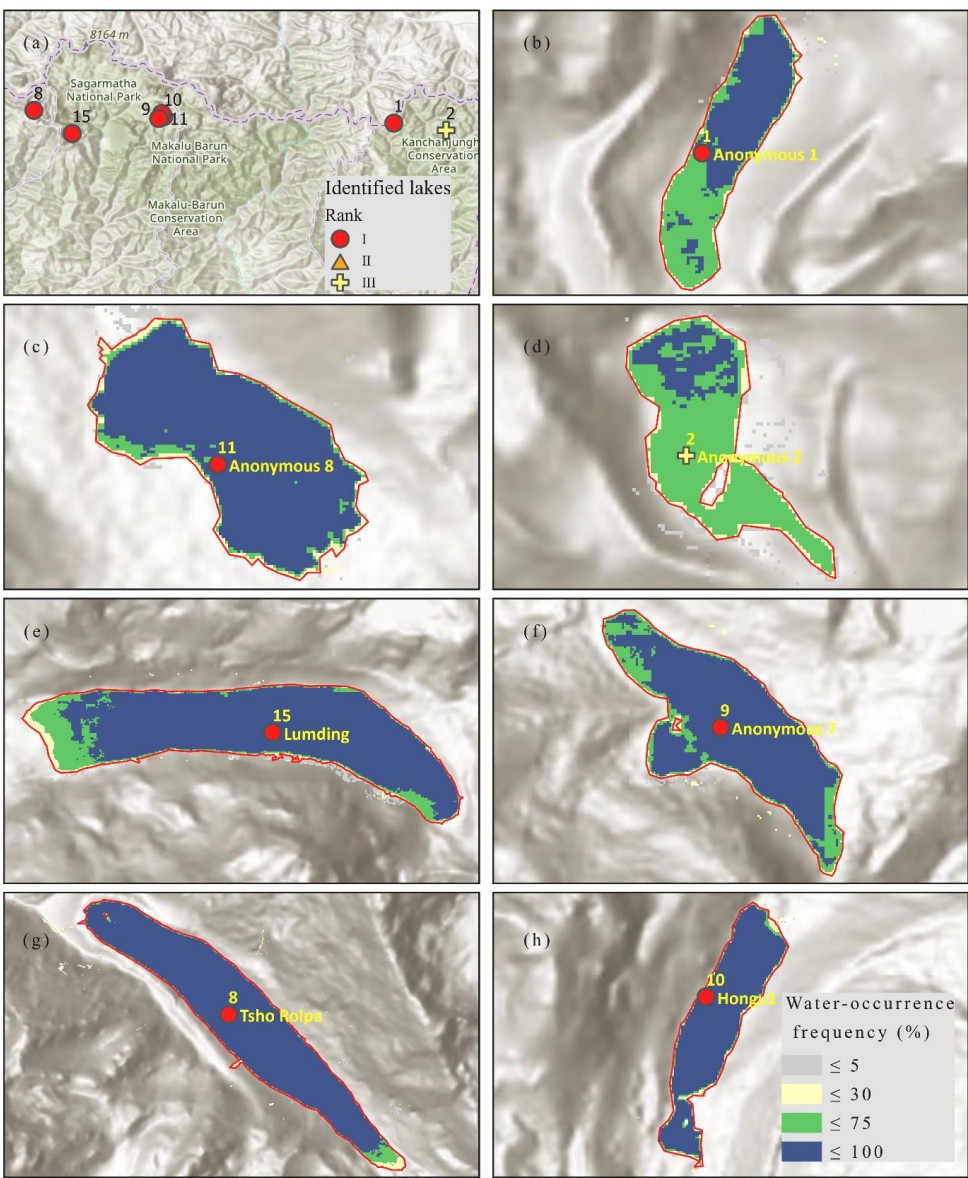

**Fig 3 Water surfaces extracted from multi-temporal Sentinel-2 imagery in representative glacial lakes in Nepal (lake numbers and other lake details can be found in Table 1)**

### 4.2 Lake volumes and peak discharges prediction

After deriving the lake area for each PDGL, we determined the posterior predicted lake depth $d$ for each lake using the Bayesian robust linear regression model developed by Veh & Walz (2020), which establishes the relationship between lake areas and





maximum depths. In this process, we obtained 100 estimates of the posterior predicted $d$, and we considered estimates falling within the 95% HDI of credible lake depth values, resulting in 94 samples of $d$ for each lake. The delineated glacial lakes from satellite imagery are assumed to have a circular shape, and we assume each glacial lake has an ellipsoidal bathymetry.

Consequently, we obtained 94 estimates of the total volume $V_{tot}$ for each lake (as shown in Fig 4 (a)). To account for the most severe GLOFs, we assume that the entire total lake volume $V_{tot}$ would be released to create GLOFs. For each of the 21 PDGLs, we predicted the peak discharge $Q_p$ based on a given value of $V_{tot}$ and η using the Bayesian piecewise linear regression model developed by Veh & Walz (2020). We generated 100 estimates of the posterior predicted $Q_p$ for each given value of $V_{tot}$ and η. The values of η for individual lakes encompass the assumed flood volumes, and we also considered 100 physically plausible

values of the breach rate $k$ based on a log-normal fit to reported breach rates. By multiplying the 94 samples of $V_{tot}$ with the 100 samples of $k$ and 100 samples of $Q_p$, we ultimately obtained a total of 940,000 scenarios of $Q_p$ per lake (as depicted in Fig 4 (b)).

The average lake volumes and peak discharges of the 21 PDGLs span more than 2 and 3 orders of magnitude. We collected geophysical investigation data for named PDGLs and compared them against calculated volumes using field-investigated lake

areas, as shown in Table 2. While there are some inconsistencies, the calculated volumes generally align with the investigated values. For example, the Lower Barun glacial lake has an average estimated flood volume of $238.9 \times 10^6$ m³ and an average estimated peak discharge of 18,240 m³/s. The water volume of the Lower Barun glacial lake in 2015 is approximately $112.3 \times 10^6$ m³, with a lake area of 1.52 km² based on bathymetric measurements. Using the established relationship between lake area and volume, the average volume for a lake with a 1.52 km² area is calculated to be $108.27 \times 10^6$ m³, which closely matches

the measured volume of the Lower Barun glacial lake. For the smallest lake (Anonymous 5) among these 21 PDGLs, its average volume and peak discharge are $0.22 \times 10^6$ m³ and 167 m³/s, respectively. This means that the average volume and peak discharge of the Lower Barun glacial lake are 1,041 and 108 times greater than those of the smallest lake, respectively.

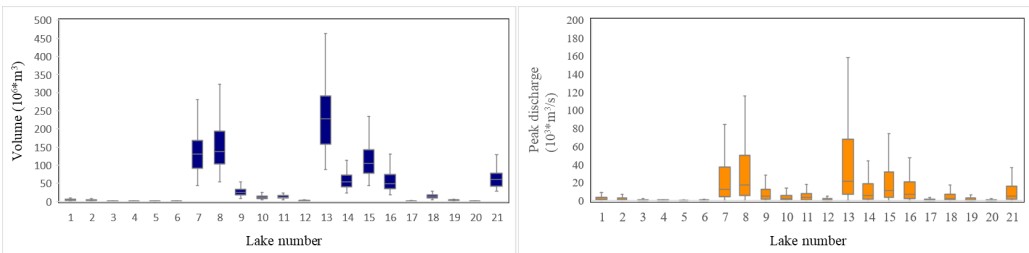

**Fig 4 (a) Estimated lake volumes and (b) estimated peak discharges for each glacial lake**





**Table 2 Comparisons between the lake areas (km²) and volumes (10⁶*m³) derived from bathymetric investigations and those calculated in this study for named lakes.**

| Lake number | Lake name | Maximum areas | Estimated volume | Investigation year | Investigated areas | Investigated volume | Calculated volume for the investigated areas | Reference |
|---|---|---|---|---|---|---|---|---|
| 7 | Imja Tsho | 1.741 | 131.16 | 2016 | 1.35 | 88 | 87.57 | Lala et al., (2017) |
| 8 | Tsho Rolpa | 1.712 | 138.39 | 1994 | 1.39 | 76.45 | 92.11 | Rana et al., (2000) |
| 13 | Lower Barun | 2.193 | 238.86 | 2015 | 1.52 | 112.3 | 108.27 | Haritashya et al., (2018) |
| 15 | Lumding | 1.475 | 103.16 | 2015 | 1.13 | 57.7 | 65.93 | Rounce et al., (2016) |
| 16 | Chamlang | 0.921 | 49.53 | 2009 | 0.87 | 34.9 - 35.6 | 45.75 | Lamsal et al., (2016) |
| 21 | Thulagi | 0.997 | 59.69 | 2017 | 0.89 | 36 | 47.12 | Haritashya et al., (2018) |





### 4.3 Flood inundation simulation

### 4.3.1 Inundation areas

Considering the substantial computational resources required for hydrodynamic simulations, 1,000 scenarios are randomly
selected from the total of 940,000 $Q_p$ scenarios per lake. In these simulations, the dam breach hydrograph is assumed to have an isosceles triangle shape, simplifying its derivation from $Q_p$ and $V_0$. The breach hydrograph then serves as the boundary conditions for the HiPIMS, which operates on a multi-GPU and multi-node high-performance programming framework to simulate GLOFs. The resultant flood dynamics provide the essential data for the subsequent evaluation of potential GLOF exposures and damages.

Herein we use the simulation results from Imja Tsho Lake and Tsho Rolpa Lake as illustrative examples to demonstrate the information obtained from the simulation results of the 1,000 GLOF scenarios. Fig. 5 reveals the flood inundation frequency and median of maximum water depth in areas with flood inundation frequency exceeding 10%. The areas with high flood inundation frequency are predominantly distributed along the downstream valley. The areas with flood inundation frequency surpassing 50% can be substantial, reaching 81.29 km$^2$ for Lower Barun lake and 185.08 km$^2$ for Tsho Rolpa lake. The median
of maximum water depth offers spatial insights into the potential severity of GLOFs on downstream areas (Fig. 5(c) and 5(d)). It facilitates the identification of areas characterized by both high inundation frequency and significant maximum water depth. For instance, concerning the Lower Barun lake, there are 14.55 km$^2$ of areas exhibiting both inundation frequency exceeding 90% and maximum water depth surpassing 1 meter. These specific areas should undoubtedly receive heightened attention in future flood risk management and mitigation.





**Fig 5 GLOF inundation frequency for (a) Imja Tsho Lake and (b) Lower Barun Lake, and median of maximum water depth for (c) Imja Tsho Lake and (d) Lower Barun Lake under respective worst situation i.e., all lake water will be released. (The basemaps used were accessed from ArcGIS Online Basemap provided by Esri.)**

The resulting inundation areas are summarized in Fig. 6 and Table 3. The median inundation extent resulting from GLOFs originating from the 21 PDGLs ranges from 2.8 km² to 190.3 km². Notably, the largest, Lower Barun glacial lake, has a median inundation area of 190.3 km², with a 95% confidence interval (CI) spanning from 3.4 km² to 315.9 km². Following closely is





Tsho Rolpa glacial lake, which faces a median inundation area of 122.0 km² (95% CI 3.8 - 231.5 km²), whereas the Imja Tsho Lake, despite having a similar lake area, anticipates a median inundation extent of 85.2 km² (95% CI 2.2 - 180.2 km²). It's worth noting that lakes that have not been extensively studied can potentially cause large inundation areas of over 10 km²,

including Anonymous 7 (86°55′41″ E, 27°51′00″ N), Anonymous 11 (86°51′29″ E, 27°41′13″ N), Anonymous 12 (85°37′51″ E, 28°09′44″ N), Anonymous 1 (87°44′59″ E, 27°48′57″ N), and Anonymous 2 (87°56′05″ E, 27°47′26″ N). The smallest lake, Anonymous 5 (87°35′46″ E, 27°42′18″ N), has a median inundation area of 2.8 km² (95% CI 1.6 - 5.3 km²). While there is a positive correlation between inundation extent and lake area (Fig 6(b)), it's important to note that inundation propagation and extent also depend on dam breach processes, as well as the underlying topography and land surface conditions of downstream

areas (Worni et al., 2012; Ancey et al., 2019). Particularly, steep and narrow valley gorges can influence flood waves, causing them to rapidly spread over long distances, often accompanied by significant physical processes such as erosion and the transport of ice, sediment, and debris.

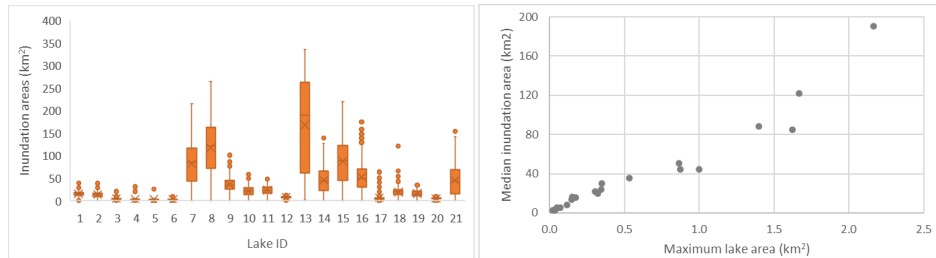

**Fig 6 (a) Simulated median inundation areas for each glacial lake (b) Relationship between the median inundation areas and lake**
**areas**





**Table 3 GLOF induced inundation areas and damage extents to buildings, roads, and agricultural lands**

| Lake number | Lake name | Inundation area (km²) | Building number | | | Building area (m²) | | | Road (km) | | | Agriculture land (km²) | | |
|---|---|---|---|---|---|---|---|---|---|---|---|---|---|---|
| | | | Slight | Moderate | Substantial | Slight | Moderate | Substantial | Slight | Moderate | Substantial | Slight | Moderate | Substantial |
| 1 | Anonymous 1 | 15.38 | 6 | 86 | 57 | 342 | 5145 | 2558 | 1.9 | 1.8 | 47.6 | 0.14 | 0.22 | 5.04 |
| 2 | Anonymous 2 | 13.92 | 5 | 76 | 54 | 276 | 4123 | 3298 | 2.1 | 1.9 | 44.0 | 0.15 | 0.23 | 3.39 |
| 3 | Anonymous 3 | 4.82 | 3 | 16 | 13 | 243 | 1208 | 850 | 1.5 | 1.4 | 15.9 | 0.00 | 0.00 | 0.02 |
| 4 | Anonymous 4 | 3.54 | 0 | 8 | 9 | 0 | 383 | 404 | 1.4 | 1.1 | 10.8 | 0.00 | 0.00 | 0.00 |
| 5 | Anonymous 5 | 2.83 | 0 | 8 | 4 | 0 | 383 | 151 | 0.8 | 0.4 | 2.2 | 0.00 | 0.00 | 0.00 |
| 6 | Anonymous 6 | 2.78 | 0 | 2 | 2 | 0 | 71 | 58 | 1.6 | 1.4 | 12.3 | 0.01 | 0.01 | 0.05 |
| 7 | Imja Tsho | 85.16 | 16 | 202 | 1295 | 1067 | 12359 | 79789 | 3.1 | 2.5 | 196.8 | 0.22 | 0.34 | 26.52 |
| 8 | Tsho Rolpa | 122.01 | 41 | 432 | 7193 | 2160 | 22463 | 401593 | 4.8 | 3.9 | 548.0 | 0.38 | 0.52 | 59.97 |
| 9 | Anonymous 7 | 35.52 | 3 | 52 | 193 | 129 | 2323 | 10685 | 1.6 | 1.1 | 46.4 | 0.10 | 0.16 | 9.36 |
| 10 | Hongu 1 | 21.79 | 1 | 40 | 28 | 34 | 1387 | 737 | 1.1 | 0.8 | 20.0 | 0.09 | 0.15 | 4.65 |
| 11 | Anonymous 8 | 23.66 | 1 | 38 | 25 | 30 | 1305 | 655 | 1.1 | 0.8 | 20.4 | 0.09 | 0.15 | 4.55 |
| 12 | Anonymous 9 | 8.06 | 0 | 3 | 5 | 0 | 113 | 287 | 1.9 | 1.0 | 17.0 | 0.00 | 0.00 | 0.00 |
| 13 | Lower Barun | 190.32 | 29 | 828 | 2770 | 1752 | 47187 | 166658 | 3.5 | 3.1 | 323.0 | 0.28 | 0.43 | 68.02 |
| 14 | Hongu 2 | 44.51 | 7 | 94 | 307 | 237 | 3527 | 15628 | 1.4 | 1.2 | 58.0 | 0.13 | 0.20 | 12.46 |
| 15 | Lumding | 88.13 | 11 | 123 | 984 | 398 | 4847 | 42101 | 1.8 | 1.7 | 170.0 | 0.22 | 0.34 | 30.40 |
| 16 | Chamling | 50.57 | 8 | 112 | 450 | 307 | 4135 | 20874 | 1.7 | 1.3 | 72.5 | 0.17 | 0.23 | 14.91 |
| 17 | Anonymous 10 | 5.35 | 0 | 3 | 7 | 0 | 47 | 108 | 0.1 | 0.0 | 1.0 | 0.00 | 0.01 | 0.13 |
| 18 | Anonymous 11 | 19.54 | 1 | 46 | 42 | 46 | 1839 | 2681 | 0.6 | 0.5 | 18.5 | 0.09 | 0.15 | 6.30 |
| 19 | Anonymous 12 | 16.21 | 61 | 462 | 315 | 4573 | 36751 | 29973 | 4.9 | 3.8 | 61.8 | 0.26 | 0.31 | 3.73 |
| 20 | Anonymous 13 | 5.1 | 0 | 0 | 0 | 0 | 0 | 0 | 1.3 | 0.7 | 6.3 | 0.00 | 0.00 | 0.00 |
| 21 | Thulagi | 44.5 | 100 | 1216 | 2421 | 7040 | 89333 | 191230 | 6.6 | 5.8 | 202.0 | 0.45 | 0.66 | 21.54 |





### 4.3.2 Exposure assessment

The exposure of objects can be spatially determined by overlaying the predicted flood inundation maps with relevant datasets
detailing buildings, roads, and agricultural land (Fig 7). The median number of inundated buildings varies from 0 to 7,792.
Out of the 21 PDGLs, 12 lakes have a median number of inundated buildings exceeding 100, while 6 of them inundate at least
1,000 buildings. The three lakes with the highest median number of inundated buildings are Tsho Rolpa, Lower Barun, and
Thulagi, each of which could inundate more than 3,000 buildings and cover an area of 2.5 * $10^5$ m² of building areas. The
number of buildings inundated by Tsho Rolpa Lake is almost twice that of Lower Barun Lake and Thulagi Lake, and the
affected areas are approximately 1.5 times larger. Overall, these well-study lakes could impact more buildings than anonymous
lakes. These 13 anonymous lakes typically affect fewer than 200 buildings, with the exceptions being Anonymous 7 (86°55′41″
E, 27°51′00″ N) and Anonymous 12 (85°37′51″ E, 28°09′44″ N), which can influence 246 and 834 buildings, respectively.
Further investigation and research are required for the two anonymous lakes. Conversely, three lakes, including Anonymous
13 (82°40′27″ E, 29°48′09″ N), Anonymous 6 (87°53′36″ E, 27°41′41″ N), and Anonymous 9 (87°05′42″ E, 27°49′44″ N),
pose lower risks, with a median number of less than 10 buildings affected.

Regarding inundated roads, the median value ranges from 1.1 to 556.8 km. Tsho Rolpa Lake, Lower Barun Lake, and Thulagi
Lake still hold the top three positions with the largest lengths of inundated roads, each exceeding 200 km. To illustrate, Tsho
Rolpa Lake, the top one in this category, has a median of 556.8 km with a 95% CI ranging from 17.0 to 876.2 km. Following
closely is Lower Barun Lake, which has inundated roads with a median of 333.9 km and a 95% CI of 9.6 to 469.4 km.
Agriculture is a cornerstone of the Nepalese economy and it is susceptible to the impacts of GLOFs. It is anticipated that eight
lakes have a median of more than 10 km² of inundated agricultural land, while four lakes have negligible impact on agriculture.
Lower Barun Lake, Tsho Rolpa Lake, and Chamlang Lake are the most perilous lakes concerning the inundation of agricultural
lands.

In addition to the high potential for human settlements to be exposed to GLOFs, hydropower projects are increasingly
vulnerable to these events. Hydropower development in Nepal has grown rapidly but unevenly. This development trend
involves projects moving upstream, bringing hydropower plants closer to glacial lakes. According to the hydropower
development data collected in the Hydro Map project (Niti Foundation, n.d.), Nepal has a total of 572 hydropower projects.
These projects include 81 that are currently operational, 180 with issued generation licenses, and 311 with issued survey
licenses. Among these, 12 existing hydropower plants (including those in operation and under construction, Table 4) are
situated close to GLOF flow channels and are potentially at risk from GLOFs due to 21 PDGLs. The 12 hydropower plants
facing such risks are Khimti I, Upper Tamakoshi, Chatara, Devighat, Trishuli, Marsyangdi, Middle Marsyangdi, Upper
Marsyangdi A, Tallo Khare Khola, Arun III, Upper Trishuli-1, and Middle Tamor. Additionally, 38 hydropower plants, for
which generation or survey licenses have been issued, are also exposed to the risk of GLOFs from these 21 PDGLs. These
hydropower plants deserve increased attention in future GLOF risk management due to their significant importance and high
vulnerability. Specifically focusing on certain lakes, Tsho Rolpa, Thulagi, and Lower Barun are responsible for potentially
inundating 6 plants (3 existing and 3 with licenses), 6 plants (3 existing and 3 with licenses), and 5 plants (2 existing and 3
with licenses), respectively. Furthermore, Lumding and Imja Tsho can each impact 4 hydropower plants with licenses.
Surprisingly, lakes Anonymous 12 (85°37′51″ E, 28°09′44″ N), Anonymous 1 (87°44′59″ E, 27°48′57″ N), and Anonymous
2 (87°56′05″ E, 27°47′26″ N) have the potential to cause the inundation of 10 plants (3 existing and 7 with licenses), 8 plants
(1 existing and 7 with licenses), and 6 plants (1 existing and 5 with licenses), respectively.



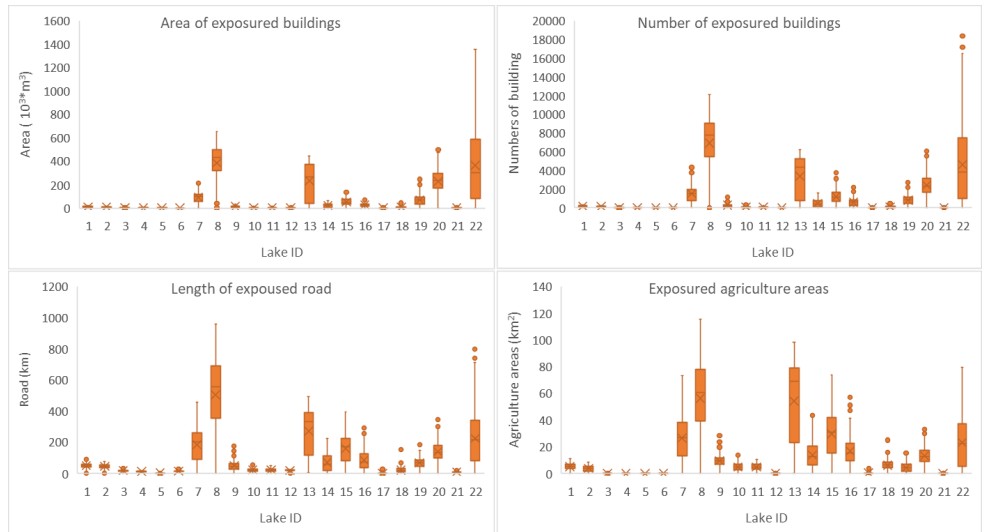

**Fig 7 Exposure objects for each glacial lake**

**Table 4 GLOF induced inundation hydropower plants**

| Hydropower plant | State | Lake name |
|---|---|---|
| Khimti I | Operation | Tsho Rolpa |
| Upper Tamakoshi | Operation | Tsho Rolpa |
| Chatara | Operation | Lower Barun |
| Devighat | Operation | Anonymous 12 |
| Trishuli | Operation | Anonymous 12 |
| Marsyangdi | Operation | Thulagi |
| Middle Marsyangdi | Operation | Thulagi |
| Upper Marsyangdi A | Operation | Thulagi |
| Tallo Khare khola | Under construction | Tsho Rolpa |
| Arun III | Under construction | Lower Barun |
| Upper Trishuli-1 | Under construction | Anonymous 12 |
| Middle Tamor | Under construction | Anonymous 1 & 2 |
| Lower Khare | Generation | Tsho Rolpa |
| Tamakoshi V | Generation | Tsho Rolpa |
| Langtang Khola Small | Generation | Anonymous 12 |
| Upper Trishuli 3A | Generation | Anonymous 12 |
| Upper Trishuli 3B | Generation | Anonymous 12 |
| Marsyangdi Besi | Generation | Thulagi |
| Upper Tamor | Generation | Anonymous 1 & 2 |
| Upper Tamor A | Survey | Anonymous 1 |
| Dudhkoshi 10 | Survey | Imja Tsho |
| Dudhkoshi-9 | Survey | Imja Tsho |
| Rolwaling Khola | Survey | Tsho Rolpa |
| Lower Isuwa Khola | Survey | Lower Barun |
| Lower Bom Khola | Survey | Lumding |
| Luja Khola | Survey | Lumding |
| Super Inkhu Khola | Survey | Anonymous 11 |
| Upper Inkhu Khola | Survey | Anonymous 11 |



| | | |
|---|---|---|
| Bhotekoshi Khola | Survey | Anonymous 12 |
| Lantang Khola Reservoir | Survey | Anonymous 12 |
| Mathillo Langtang | Survey | Anonymous 12 |
| Upper Trishuli-I Cascade | Survey | Anonymous 12 |
| Rigdi Khola | Survey | Thulagi |
| Dana Khola | Survey | Thulagi |
| Tamor Mewa | Survey | Anonymous 1 & 2 |
| Tamor Khola-5 | Survey | Anonymous 1 & 2 & 3 & 6 |
| Ghunsa-Tamor | Survey | Anonymous 1 & 6 |
| Super Tamor | Survey | Anonymous 1 & 6 |
| Upper Tamor HEP | Survey | Anonymous 1 & 6 |
| Lower Barun Khola | Survey | Lower Barun & Anonymous 9 |
| Upper Barunkhola | Survey | Lower Barun & Anonymous 9 |
| Ghunsa Khola | Survey | Anonymous 2 & 3 |
| Ghunsa Khola | Survey | Anonymous 2 & 3 |
| Chujung Khola | Survey | Anonymous 4 & 5 |
| Dudhkoshi-6 | Survey | Imja Tsho, Lumding |
| Surke Dudhkoshi | Survey | Imja Tsho, Lumding |
| Hongu Khola I | Survey | Hongu 1& 2, Chamlang, Anonymous 7, 8 & 10 |
| Middle Hongu Khola B | Survey | Hongu 1& 2, Chamlang, Anonymous 7, 8 & 10 |
| Middle Hongukhola A | Survey | Hongu 1& 2, Chamlang, Anonymous 7, 8 & 10 |
| Hongu Khola | Survey | Hongu 1& 2, Chamlang, Anonymous 7, 8 & 10 |

### 4.3.3 Damage Assessment

GLOF damage assessment relies on spatial inundation maps of water depth and depth-damage curves. The inundation maps, depicting water depth, are represented by the median of maximum water depths under various scenarios. In accordance with the technical manual of the HAZUS Flood model (FEMA, 2009), damage extents of 1% to 10%, 11% to 50%, and 50% to 100% are defined as slight, moderate, and substantial damage, respectively. Table 3 provides estimates of damage for buildings, roads, and agricultural lands for each lake. In the case of Tsho Rolpa Lake, approximately 7,193 buildings are

projected to suffer substantial damage from GLOFs. Similarly, Lower Barun Lake and Thulagi Lake are expected to cause substantial damage to approximately 2,500 buildings each. Other lakes, such as Imja Tsho Lake and Lumding Lake, are estimated to impact roughly 1,000 buildings with substantial damage. Notably, Anonymous 12 (85°37′51″ E, 28°09′44″ N) has the potential to affect 838 buildings, with 462 experiencing moderate impact and 315 facing substantial damage. Situated in the Trishuli River Basin, Anonymous 12 not only faces a high hazard level (Rank I) but also high exposure. On the other

hand, another anonymous lake (Anonymous 13, at 82°40′27″ E, 29°48′09″ N) faces a relatively high hazard level (Rank II) but is not projected to inundate any buildings due to GLOFs. For PDGLs with a high number of impacted buildings (more than 1,000), except for Thulagi Lake, more than 75% of the impacted buildings are expected to incur substantial damage. In all PDGLs, most affected buildings (over 90%) are predicted to experience moderate or substantial damage. Likewise, nearly all exposed roads and agricultural lands are anticipated to undergo substantial damage due to high levels of maximum water

depths.

### 4.3.4 Sensitivity analysis

To account for all possible glacial lake outburst scenarios, less severe conditions are also considered, where 25% and 50% of the lake water volume is released. In each of these less severe scenarios, 100 cases are randomly selected from a total of 940,000 samples. The outcomes of these scenarios will be compared to the worst-case condition, which has been presented

earlier. Fig 8 illustrates the median values for inundation area, the number of inundated buildings, the length of inundated



roads, and inundated agricultural areas resulting from GLOFs. In the case of Lower Barun Lake, the release of 25% and 50% of the lake water leads to the inundation of 74 km² and 159 km² of downstream areas, respectively. When 100% of the lake water is released, the inundation areas are 2.06 and 1.11 times larger than those under the 25% and 50% lake water release scenarios, respectively. Following Lower Barun Lake, Tsho Rolpa Lake and Imja Tsho Lake have the potential to cause

significant inundation areas. Even with just 25% of the lake water being released, Tsho Rolpa Lake and Imja Tsho Lake can potentially submerge approximately 50 km² of areas. In terms of the potential impacts on buildings and roads, Tsho Rolpa Lake, Lower Barun Lake, and Thulagi Lake are the top three lakes that could experience the most significant damages. Especially for Tsho Rolpa Lake, the most severely affected, a 25% lake water release could impact 4,061 buildings and 271 km of roads, while a 50% lake water release could inundate 7,096 buildings and 494 km of roads. When it comes to agricultural

areas, Tsho Rolpa Lake, Lower Barun Lake, and Lumding Lake are likely to sustain the most damage.

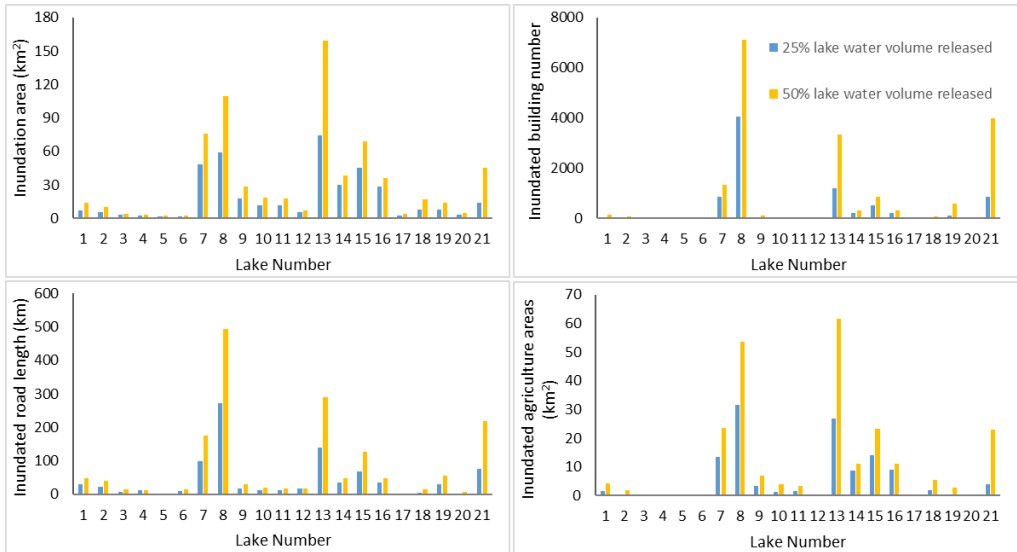

**Fig 8 Median values of inundation area, inundated building number, inundated road length and inundated agricultural areas due to GLOFs under 25% and 50% of lake water volume released**

## 5 Discussion

The results of this study showcase how object-based GLOF exposure and impact assessments can be carried out for multiple glacial lakes across the data-scarce Himalayan region on a national scale, leveraging established techniques and methods. In doing so, this study relies on several key components, including remote sensing techniques for accurate glacial lake area delineation, Bayesian regression models for deriving lake water depth and peak discharge relationships (Veh & Walz, 2020), state-of-the-art flood modelling technology supported by parallelized high-performance computing (Zhao et al., 2022), and

object-based GLOF exposure and impact evaluation using open-source data (Chen et al., 2022). Open data and images from various sources are harnessed to generate input data for flood modelling and object-based exposure datasets, addressing the challenges posed by limited data availability and the inaccessibility of many glacial lakes in high-altitude regions. Recognizing the potential impact of small sample sizes and outliers in developing relationships between glacial lake volumes and peak discharges, a Bayesian approach is employed to derive plausible value ranges for lake volumes and peak discharges. This

approach allows for the creation of multiple possible GLOF scenarios for each glacial lake. High-resolution inundation simulations for various GLOF scenarios are conducted using flood modelling technology, supported by parallelized high-performance computing, facilitating subsequent object-based assessments of GLOF exposures and impacts. This





comprehensive methodology framework is applied to assess hazardous glacial lakes in Nepal, identified as having a high likelihood of outburst floods. The results offer a complete, clear, and detailed understanding of potential exposures and impacts
stemming from these PDGLs. While much of the prior work has concentrated on the initial step of GLOF risk evaluation, specifically hazard assessment for glacial lakes, this study advances the field by addressing the second stage, which involves exposure and impact evaluation. The insights gained from this study can empower authorities with not only knowledge of where threats exist but also an understanding of the expected magnitude of impacts. This study aligns with and contributes to Nepal's national strategy in disaster risk reduction. For instance, it supports objectives such as "increasing understanding of
disaster risk and ensuring access to related information at all levels," as highlighted in the National Policy for Disaster Risk Reduction. Additionally, it aligns with strategic activity of "assessing geo-referenced flood exposure and vulnerability for flood-prone infrastructure," as highlighted in the Disaster Risk Reduction National Strategic Action Plan 2018-2030.

Glacial lakes are mostly situated in remote, hard-to-reach areas, which makes conducting detailed bathymetric surveys challenging. As a result, unmeasured lakes often require estimates of their depths and volumes using empirical relationships
derived from bathymetric datasets (e.g., Huggel et al., 2002; Kapitsa et al., 2017). Similarly, potential peak discharges of outburst floods from these glacial lakes are usually estimated using either empirical relationships (e.g., Huggel et al., 2002) or physics-based models (e.g., Walder & O'Connor, 1997). However, it's worth noting that the estimated lake volumes and potential peak discharges derived from existing empirical relationships can vary significantly, sometimes by up to an order of magnitude (Cook and Quincey, 2015; Muñoz et al., 2020). To account for the uncertainty associated with these estimated lake
volumes and potential peak discharges, we have employed Bayesian regression models that establish relationships between lake areas and depths, as well as between flood volume and peak discharge (Veh & Walz, 2020). The Bayesian approach allows us to quantify uncertainties related to models and parameters by simultaneously evaluating the variability and uncertainty within the observational data, going beyond classical frequentist methods (Ellison, 2004). This Bayesian treatment enables us to make predictions by integrating over the distribution of model parameters, rather than relying on specific
estimated parameters (Bishop & Tipping, 2003). Consequently, predictive posteriors of lake depths and peak discharges for each lake are generated and curtailed within the 95% HDI. This provides a plausible range of values for lake volumes and peak discharges for each glacial lake, as opposed to offering a single value estimate, thus enabling a more objective and detailed assessment of downstream impacts of PDGLs in Nepal.

GLOFs can have a significant impact due to the large volume of water stored in glacial lakes, resulting in rapid breaches and
high outflow peaks and total discharges. Among the 21 PDGLs in Nepal, Tsho Rolpa Lake, Lower Barun Lake, and Thulagi Lake are expected to experience the most severe impacts of GLOFs on buildings and roads, while Lower Barun Lake, Tsho Rolpa Lake, and Lumding Lake are anticipated to be the most impacted in terms of GLOFs on agricultural areas. Rounce et al. (2016, 2017) also assessed the downstream impacts of GLOFs from glacial lakes in the Nepal Himalaya. They likewise identified Tsho Rolpa Lake, Lower Barun Lake, and Thulagi Lake as having the most affected buildings, while two anonymous
lakes and Thulagi Lake were anticipated to experience the most significant impacts on agricultural areas. However, it's important to note that Rounce et al. (2016, 2017) employed the Monte Carlo Least Cost Path model (MC-LCP, Watson et al., 2015) to estimate the extent of GLOFs for each lake. While the MC-LCP model is computationally efficient and suitable for large-scale applications, it lacks a physical basis and relies solely on the terrain conditions downstream along the river channel, without considering variations in lake release volumes and peak discharges. As a result, flood extents for lakes with differing
potential flood volumes may be indistinguishable. Another limitation is that the threshold for the cut-off distance in MC-LCP needs to be artificially set, while the realistic cutoff distance downstream for each lake varies, sometimes extending over 200 km downstream (Richardson & Reynolds, 2000). This study takes a different approach by employing a physics-based hydrodynamic model, which predicts not only inundation extent but also the spatial characteristics of flood features, including





inundation frequency and water depths, while considering various outburst scenarios. This information can be used to identify

potential exposures and assess the extent of damages to which exposures may be subject.

In addition to the growing vulnerability of human settlements in mountainous regions, there is an increasing exposure of infrastructure related to energy security and commerce to GLOFs. Therefore, an objective assessment of the risk to infrastructure posed by PDGLs is crucial. This study considers hydropower plants, given their critical importance and rapid development in Nepal. Nepal is at the heart of a modern resurgence in hydropower development in the Himalayas (Lord et al.,

2016). The country boasts abundant hydropower resources thanks to its ample river water, steep gradients, and mountainous terrain. In fact, Nepal has the potential to generate over 90,000 megawatts of hydropower, with at least 42,000 megawatts considered technically and economically feasible from its three major river systems and their smaller tributaries (Alam et al., 2017). Despite this rich hydropower potential, Nepal currently generates only around 847 megawatts from its hydropower resources. This is coupled with a significant energy scarcity issue, with nearly half of Nepal's population lacking access to

grid-connected power. Nepal is driven by a strong ambition to become a 'hydropower nation', as evidenced by ongoing and intensifying efforts in hydropower development. At present, a considerable number of hydropower projects are in the planning and construction stages (46 projects exceeding 100 gigawatts) to enhance the country's overall generating capacity. These planned hydropower projects are primarily situated along rivers connected to glaciers located in the northern region of Nepal (Shakti et al., 2021). While a few existing hydropower plants have experienced direct impacts from recorded GLOFs, such as

the Namche hydroelectric power plant destroyed by the 1985 Dig Tsho GLOF (Vuichard & Zimmermann, 1987) and the Bhotekoshi hydropower plant affected by the 2016 GLOF (Cook et al., 2018), GLOFs can be highly destructive and unpredictable, posing a significant threat to hydropower facilities. Furthermore, the expansion of hydropower plants into the upstream regions of watersheds substantially increases the vulnerability of infrastructure to GLOFs (Nie et al., 2021). Schwanghart et al. (2016) estimated that two-thirds of the existing and planned hydropower projects in the Himalayas are

located in areas potentially affected by GLOFs, and up to one-third of these projects could face GLOF discharges exceeding their local design flood capacities. In this study, we have identified that 50 existing and planned hydropower projects could potentially be impacted by GLOFs originating from 21 PDGLs. We strongly urge stakeholders responsible for planning, designing, constructing, and managing infrastructure to take these GLOF risks into consideration. It is crucial to develop proactive adaptation measures and adopt sustainable solutions to minimize the negative impacts of GLOFs on infrastructure.

In addition to well-studied PDGLs like Tsho Rolpa Lake, Thulagi Lake, and Lower Barun Lake, some anonymous lakes also present a significant risk of GLOFs. For instance, Anonymous 12, 7, 1, and 2 pose high GLOF risks. Anonymous 12 (85°37′51″ E, 28°09′44″ N; 4990m above sea level), Anonymous 7 (86°55′41″ E, 27°51′00″ N; 5406m above sea level), and Anonymous 1 (87°44′59″ E, 27°48′57″ N; 4880m above sea level) are categorized as Rank I PDGLs, while Anonymous 2 (87°56′05″ N, 27°47′26″ E; 4950m above sea level) falls into the Rank III category. GLOFs from any of these four lakes have the potential

to impact more than 100 buildings, and GLOFs resulting from Lakes Anonymous 12, 1, and 2 may submerge existing hydropower facilities. Unfortunately, there is limited information available about these anonymous lakes in comparison to well-studied PDGLs. To gain a better understanding of their conditions, a comprehensive research strategy is needed, which includes fieldwork investigations, remote sensing techniques, and modelling approaches. This study has leveraged remote sensing techniques and modelling approaches to preliminarily identify PDGLs with a high level of exposure and potential

impacts from GLOFs. However, it is imperative to conduct fieldwork investigations, including in situ measurements, to obtain the essential information required to comprehend the actual state of these anonymous lakes at the local scale. These field investigations will also serve as ground truthing to calibrate remote sensing-based data and model outputs. Moreover, considering the challenging nature of fieldwork in glacial lake areas, the cost of expeditions, and the high level of fitness and expertise required by monitoring teams, the preliminary identification of PDGLs with high exposure and potential impacts can

offer valuable evidence to support decision-making in the allocation of financial and human resources.



We acknowledge the importance of validating the proposed framework for estimating the impact of GLOFs while recognizing the inherent challenges associated with validation due to the limited availability of historical data. Although Nie et al. (2018), Veh et al. (2019), and Shrestha et al. (2023) have provided valuable inventories of historical GLOFs in the Himalayas, these primarily provide information on the date and location of outbursts, offering limited or no information on the actual impacts

resulting from historical GLOFs. Even when impact data is available, it often comprises only generalized descriptions, encompassing metrics like the overall number of casualties, infrastructure damage, and affected villages, lacking specific spatial information. Consequently, obtaining adequate data for validating our proposed impact estimation framework for GLOFs proves challenging. It is noteworthy that our proposed framework employs the fully physically based hydrodynamic model HiPIMS, intricately designed to capture the highly transient and complex hydrodynamic processes induced by events

such as dam breaks and flash floods. HiPIMS has been successfully validated for these extreme flow conditions (e.g., Luke and Liang, 2013; Liang et al., 2016). The adoption of this model enhances our confidence in simulating the spatial-temporal processes of GLOF inundation, ultimately contributing to improved hazard evaluation results. Furthermore, we employ Bayesian approaches to derive plausible value ranges for lake volumes and peak discharges. These approaches facilitate the creation of multiple GLOF scenarios for each glacial lake, ensuring comprehensive coverage of all potential glacial lake

outburst scenarios. The incorporation of Bayesian methods allows us to account for uncertainties, thereby enhancing the robustness of our impact evaluation for potentially devastating GLOFs.

## 6 Conclusion

Exposure and damage estimations are integral components of GLOF risk assessment. Having sufficient information about the potential impacts of GLOFs originating from PDGLs is essential to facilitate GLOF risk management. In this study, we

harnessed multi-temporal satellite imagery, Bayesian regression models that establishes relationships between lake areas and depths, as well as between flood volume and peak discharge, and a high-performance hydrodynamic flood model to support GLOF exposure and damage assessments for multiple lakes. We applied this assessment framework to 21 PDGLs identified in the Nepal Himalaya, and the key findings of this study are summarized as follows:

- Utilizing multi-temporal imagery capturing different hydrological conditions of glacial lakes enables the derivation of
the full or maximum glacial lake boundaries for potential risk assessment.

- The Bayesian regression model, which establishes relationships between lake areas and depths, as well as between flood volume and peak discharge, can produce predictive posterior distributions for lake depths and peak discharges for each lake. These distributions offer a plausible range of values for lake volumes and peak discharges for each PDGL, facilitating subsequent objective flood modelling and impact analysis.

- The hydrodynamic model (HiPIMS), supported by parallelized high-performance GPU computation, is capable of predicting the resulting GLOFs in terms of temporally and spatially varying flood frequency and water depths to reflect the highly transient flood dynamics under various scenarios for multiple glacial lakes on a large scale.

- Among the 21 PDGLs identified in the Nepal Himalayas, Tsho Rolpa Lake, Lower Barun Lake, and Thulagi Lake are poised to bear the most severe impacts of GLOFs on buildings and roads. Meanwhile, Lower Barun Lake, Tsho Rolpa
Lake, and Lumding Lake will encounter the most significant GLOF impacts on agricultural areas. Four anonymous lakes, specifically Anonymous 12 (85°37′51″ E, 28°09′44″ N; 4990m above sea level), Anonymous 7 (86°55′41″ E, 27°51′00″ N; 5406m above sea level), Anonymous 1 (87°44′59″ E, 27°48′57″ N; 4880m above sea level), and Anonymous 2 (87°56′05″ N, 27°47′26″ E; 4950m above sea level), have the potential to impact more than 100 buildings. Notably, Anonymous 12, 1, and 2 may even submerge existing hydropower facilities. The GLOFs from these 21 PDGLs also can



impact the 12 existing hydropower plants and the 38 hydropower projects that have been granted generation or survey
        licenses.

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

**Data availability**

The DEM used in this work is Shuttle Radar Topography Mission (SRTM) DEM. Land use types are extracted from the Landsat TM imagery from the year 2010, which can be accessed at

http://rds.icimod.org/Home/DataDetail?metadataId=9224.The OpenStreetMap (OSM) data can be accessed via the link http://download.geofabrik.de/asia/nepal.html. Hydropower plant data are obtained from Hydro Map project through the link https://hydro.naxa.com.np/core/about.

**Code availability**

The flood model can be accessed through the link https://github.com/HEMLab/HiPIMS-CUDA.

**Author contribution**

HC was responsible for developing the methodology, conducting analysis, and drafting the paper. QL handled funding acquisition, research design, and reviewed and refined the draft. JZ developed the flood model codes, and SM provided a review of the draft.





**Competing interests**

The contact author has declared that none of the authors has any competing interests.

**Acknowledgments**

This work is supported by the WeACT project (NE/S005919/1) funded by the UK Natural Environment Research
Council (NERC) through the SHEAR programme.