# Peer review of "Assessing national exposure and impact to glacial lake outburst floods considering uncertainty under data sparsity"

_Hydrology and Earth System Sciences, 2023_

## Author Comment (AC2)

** Please find our response highlighted in blue to the reviewer's comments.

This paper established the method for quantitative assessment of GLOF risk by combining Random Forest model to extract lake surface area, well-established Bayesian regression models to estimate glacial lake volume and peak discharge of outburst flood, hydrodynamic flood modelling, and damage analysis. Then the framework was applied to assess 21 glacial lakes in the Nepal Himalaya. I enjoy reading the methodology and believe it will contribute greatly to the quantitative assessment of the glacier lakes in the Himalaya with data sparsity. However, several major issues still need to be addressed carefully before further consideration of publication for this manuscript.

Response: Thank you so much for the reviewer's overall positive feedback.

First, an isosceles triangle shape was assumed for the hydrograph of outburst floods (line 376, this information is better to be shown in the methods section by the way). I understand this assumption would simplify the calculation of hydrograph, which acts the key input for 2D hydrodynamic model. But this assumption needs to be justified before it can be used. If the hydrological monitoring data for the hydrograph of GLOF is too scarce, the authors can check the measured hydrographs of outburst floods for glacial lakes or barrier lakes in experimental research and see whether this assumption is close to the observations. The hydrograph shapes affect the interaction between morphology and hydraulics along the river significantly, so the assumption here needs to be made very carefully.

Response: Thank you for your understanding and valuable advice. The assumption of an isosceles triangle shape for the dam breach hydrograph has been validated through experimental observations and simulation results obtained from commonly used mechanisms and empirical models. This information has been relocated to the methodology section. The revised text is presented in the methodology section of the main text and references as follows:

In these simulations, the dam breach hydrograph is assumed to have an isosceles triangle shape, simplifying its derivation from $Q_p$ and $V_0$. The breach hydrograph then serves as the boundary conditions for the hydrodynamic modelling. Although there is some uncertainty, the assumption of an isosceles triangle shape for the dam breach hydrograph aligns with experimental observations (e.g., Morris et al., 2007; Walder et al., 2015; Yang et al., 2015) and is supported by simulation results from commonly used mechanisms and empirical models (e.g., Yang et al., 2023).

Morris, M. W., Hassan, M. A. A. M., & Vaskinn, K. A. (2007). Breach formation: Field test and laboratory experiments. *Journal of Hydraulic Research*, 45(sup1), 9-17.

Walder, J. S., Iverson, R. M., Godt, J. W., Logan, M., & Solovitz, S. A. (2015). Controls on the breach geometry and flood hydrograph during overtopping of noncohesive earthen dams. *Water Resources Research*, 51(8), 6701-6724.

Yang, M., Cai, Q., Li, Z., & Yang, J. (2023). Uncertainty analysis on flood routing of embankment dam breach due to overtopping failure. *Scientific Reports*, 13(1), 20151.

Yang, Y., Cao, S. Y., Yang, K. J., & Li, W. P. (2015). Experimental study of breach process of landslide dams by overtopping and its initiation mechanisms. *Journal of Hydrodynamics*, 27(6), 872-883.

Second, I did not find the points in classifying the glacial lakes into three categories (lines 275-277). The classification standards were blurry and the glacier lakes were not analyzed by category (e.g., the volumes, peak discharges, or inundation areas of each class) in the results. I do not think it will make much difference to the clarity of the results if the classification is removed but will help reduce the length of the manuscript, which is already a bit too long.

Response: Great advice. The classification details have been removed from the main text.

Third, the manuscript is verbose in some sections and will benefit a lot if the irrelevant or repeating information is removed. For example, in lines 318 to 322, the lake areas from literature are listed, but these are not the results or findings of this study. So these lines can be shortened into one short sentence indicating the two glacier lakes are expending rapidly. Another example is the first paragraph in the discussion section. The paragraph adds very little information, mainly repeating what has been done in this work. It is fine to summarize the work in this study as the start of discussion but the summary needs to be concise. The second paragraph in the discussions has the same issue, with repeating information from the introduction and methodology section.

Response: We have shortened lines 318 to 322 into a short sentence as below. We have removed the first and second paragraphs of the discussion section to avoid redundant information with the introduction and methodology, and to shorten the length of the paper.

~~Remarkably, Lower Barun Lake has undergone significant area growth since its initial appearance, with an area of 0.04 km2 in 1987 (Sattar et al., 2021), 0.64 km2 in 1989 (Maskey et al., 2020), 1.79 km2 in 2017 (Haritashya et al., 2018), 2 km2 in 2018 (Maskey et al., 2020), and 2.09 km2 in 2019 (Sattar et al., 2021). Imja Tsho Lake, the second largest PDGL, also underwent rapid growth in both area and volume. It did not exist in 1960, but its area in 1963, 1992, 2002, and 2012 measured 0.03, 0.648, 0.868, and 1.257 km2, respectively (Budhathoki et al., 2010; Somos-Valenzuela et al., 2014).'~~ → Lower Barun Lake, along with the second largest PDGL, Imja Tsho Lake, has undergone significant area growth.

Apart from being verbose, the discussion section needs to be more focused. In lines 551 to 566, the authors introduced the backgrounds of hydropower projects in Nepal. This may help the readers to understand why the risk of hydropower stations was evaluated in this work, but too many details may become a deviation from discussing how the risk is distributed and varying in Nepal. Such information is more proper to be put into the supplementary materials rather than the maintext.

Response: Agreeing with the reviewer's comments, we have removed lines 550-555 and only retained the most crucial information relevant to GLOF risk.

Although the discussions include some comparisons with other studies to show the advantage of the methodology, I suggest the authors work on improving the depth of the discussions. For example, the assessment of inundation, exposure and damage has been presented in the results section, but the spatial distribution pattern, key influencing factors and the reasons or mechanism for the most severely affected glacier lakes can be further discussed. The discussion on the performance of the method used in this study is already enough but the interpretations of the outcomes of the method have not been dealt with in

depth. But the interpretations will provide crucial insight to risk management of the glacier lakes for the study area.

Response: We really appreciate the valuable comments on improving the depth of analysis. However, since our focus was solely on 21 potentially dangerous glacial lakes rather than examining all glacier lakes in Nepal, it is difficult to identify the spatial distribution pattern of GLOF risk. When considering the key influencing factors and underlying mechanisms, we directly investigated the identified potentially dangerous glacial lakes, whose hazard factors have been scrutinized in existing studies. This study expands upon prior research by examining the exposure and impact situation of GLOFs, primarily influenced by downstream topography, community, and building locations. Therefore, we opted not to specifically analyse some specific lakes, avoiding the repetition of existing hazard factors or a simple description of their downstream conditions.

Last, so many abbreviations were used in the manuscript but a list of abbreviations is missing. This creates extra difficulty for the readers to follow the manuscript.

Response: Abbreviations have been checked and all are defined at the first instance in the text, and a corresponding list of abbreviations has been included for reference.

Appendix A: List of abbreviations used in this study.

| DEM | digital elevation model |
|-----|-------------------------|
| EVI | Enhanced Vegetation Index |
| GIS | Geographic Information System |
| GLOFs | Glacial Lake Outburst Floods |
| GPU | Graphics processing unit |
| HiPIMS | High-Performance Integrated Hydrodynamic Modelling System |
| MNDWI | Modified Normalized Difference Water Index |
| NIR | Near Infrared |
| NDMI | Normalized Difference Moisture Index |
| NDVI | Normalized Difference Vegetation Index |
| NDWI | Normalized Difference Water Index |
| OSM | OpenStreetMap |
| PDGL | potentially dangerous glacial lake |
| SRTM | Shuttle Radar Topography Mission |
| TOA | Top-Of-Atmosphere |

Also, figures need to be refined. Figures 4, 6, 7 and 8 do not show any ticks on the axes while the flow directions should be marked in figure 5.

Response: Ticks have been added to the axes in Figures 4, 6, and 7, shown below. Figure 8 has been removed due to the new analysis of the GLOF simulation and impact results based on the comments from the other reviewer.

[Figure]

Flow directions have been marked in Figure 5.

[Figure]

Specific comments

Line 38: revise "... has observed..." to "is experiencing".

Response: Revised according to the comment.

Line 42: change "an objective and reproducible assessment" to "the requirement for reproducible assessment".

Response: Revised according to the comment.

Line 45: remove "typically focus on individual glacial lakes, which".

Response: Revised according to the comment.

Lines 54-57: the sentence can be more concise. Please rewrite.

Response: Revised according to the comment, as below.

 → However, the complexity of GLOFs renders simple flood models inadequate for capturing their dynamics, thereby making them incapable of supporting detailed assessments of potential impacts on downstream communities and infrastructure.

Line 85: reference(s) are needed after "impact of GLOFs".

Response: Revised according to the comment, as below.

Previous studies have typically relied on census data at coarse spatial resolutions or aggregated land use data that encompass various objects like properties and infrastructure, to estimate the potential socio-economic impact of GLOFs (e.g., Shrestha & Nakagawa, 2014; Rounce et al., 2016).

Shrestha, B. B., & Nakagawa, H. (2014). Assessment of potential outburst floods from the Tsho Rolpa glacial lake in Nepal. *Natural Hazards*, 71(1), 913-936.

Rounce, D. R., McKinney, D. C., Lala, J. M., Byers, A. C., & Watson, C. S. (2016). A new remote hazard and risk assessment framework for glacial lakes in the Nepal Himalaya. *Hydrology and Earth System Sciences*, 20(9), 3455-3475.

Line 216: reference(s) are needed after "CPU-based counterpart".

Response: Revised according to the comment, as below.

It's worth noting that the GPU-accelerated model has demonstrated computational efficiency up to ten times greater than its CPU-based counterpart (Smith & Liang, 2013).

Smith, L. S., & Liang, Q. (2013). Towards a generalised GPU/CPU shallow-flow modelling tool. *Computers & Fluids*, 88, 334-343.

Line 231: the year seems to be 2022 from the reference list.

Response: The error has been rectified.

Line 239: are the values of Manning coefficients appropriate for Nepal? Please justify this setting.

Response: The Manning coefficients 0.016 to 0.15 were specified based on values provided in earlier hydraulic textbooks or reports (such as Chow, 1959; Barnes, 1967; Arcement and Schneider, 1984), aligning with previous studies, for example, 0.035 to 0.17 in Nepal (Sattar et al., 2021) and 0.035 to 0.120 in Bhutan (Rinzin et al., 2023).

Sattar, A., Haritashya, U. K., Kargel, J. S., Leonard, G. J., Shugar, D. H., & Chase, D. V. (2021). Modeling lake outburst and downstream hazard assessment of the Lower Barun Glacial Lake, Nepal Himalaya. *Journal of Hydrology*, 598, 126208.

Rinzin, S., Zhang, G., Sattar, A., Wangchuk, S., Allen, S. K., Dunning, S., & Peng, M. (2023). GLOF hazard, exposure, vulnerability, and risk assessment of potentially dangerous glacial lakes in the Bhutan Himalaya. *Journal of Hydrology*, 619, 129311.

Lines 345-357: most of the paragraph should be moved to the methods part. Please consider.

Response: The paragraph has been moved to Section 2.2.1 within the methodology, as below.

To account for the most severe GLOFs, we assume that the entire total lake volume $V_{tot}$ would be released to create GLOFs. For each lake, we predicted the peak discharge $Q_p$ based on a given value of $V_{tot}$ and η using the Bayesian piecewise linear regression model. We generated 100 estimates of the posterior predicted $Q_p$ for each given value of $V_{tot}$ and η. The values of η for individual lakes encompass the assumed flood volumes, and we also considered 100 physically plausible values of the breach rate $k$ based on a log-normal fit to reported breach rates. By multiplying the 94 samples of $V_{tot}$ with the 100 samples of $k$ and 100 samples of $Q_p$, we ultimately obtained a total of 940,000 scenarios of $Q_p$ per lake. Considering the substantial computational resources required for GLOF inundation simulations in section 2.2.2, 1,000 scenarios are randomly selected from the total of 940,000 $Q_p$ scenarios per lake.

Figure 5: The locations of inset plots in the big map need to be marked.

Response: The locations of inset plots have been marked as below.

[Figure]

Lines 402-405: The sentence should be moved to discussions.

Response:  Yes, this has been moved to discussions.

Figure 8: It may be clearer if the results for the scenario when 100% lake water is released are presented together with the less severe scenarios.

Response: We have addressed this comment by incorporating the less severe scenarios alongside the 100% lake water scenarios as shown below.

To account for all possible glacial lake outburst scenarios, less severe conditions are also considered, where 25%, 50% and 75% of the lake water volume is released. In each of these less severe scenarios, 100 cases are randomly selected from a total of 940,000 samples. The outcomes of these scenarios will be compared to the worst-case conditions. Fig 7 illustrates the inundation area for inundation probabilities exceeding 50% resulting from GLOFs. In the case of Lower Barun Lake, the release of 25% and 50% of the lake water leads to the inundation of 50.2 km$^2$ and 60.6 km$^2$ of downstream areas, respectively. When 100% of the lake water is released, the inundation areas are 1.29 and 1.08 times larger than those under the 25% and 50% lake water release scenarios, respectively. Following Lower Barun Lake, Tsho Rolpa Lake and Lumding Lake have the potential to cause significant inundation areas. Even with just 25% of the lake water being released, Tsho Rolpa Lake and Imja Tsho Lake can potentially submerge approximately 30 km$^2$ of areas for inundation probabilities exceeding 50%.

---

## Author Comment (AC3)

** Please find our response highlighted in blue to the reviewer's comments.

This study aims at modelling potential future GLOFs from 21 Nepalese glacial lakes identified as potentially dangerous in previous study by Bajracharya et al. (2020). This study uses regression model to estimate plausible ranges of mean lake depths and so lake volumes and peak discharges. Further, exposed elements within the limits of modelled inundation areas are mapped and potential damage is assessed. Undoubtedly, such results are of value for disaster risk management authorities and I appreciate holistic approach going beyond GLOF modelling itself. While I'm very much in favor of GLOF hazard / risk assessment studies that consider a range of scenarios and I really appreciate the amount of work done, I have some thoughts for further improvements in the hazard assessment part.

Response: Thank you very much for the reviewer's overall positive feedback.

Similar approach has been developed and employed by Veh et al. (2020) across the whole Himalaya. While thousands of lakes were considered in their study and the approach was suitable, I would expect bit more site-specific input data in case study of 21 lakes. For instance, moraine dam geometry (height and width) could be used to estimate max. breach depth and so max. flood volume that could differ substantially compared to the assumption of 100% lake volume release which is: (i) in general very unlikely for large lakes; and (ii) physically not even possible in many cases (when the height of a damming moraine is less than max. depth of the lake or the geometry of the dam is very flat).

Response: I believe the reviewer is referring to Veh & Walz (2020), Proceedings of the National Academy of Sciences, 117(2), 907-912. We have indeed adopted the same approach developed and employed by Veh & Walz (2020). However, in addition to hazard evaluation that Veh & Walz (2020) focused on, our focus extends to exposure and impact assessments of GLOFs. To achieve this, our study relies on several key components. These include remote sensing techniques for accurate glacial lake area delineation, Bayesian regression models for deriving relationships between lake water depth and peak discharge (Veh & Walz, 2020), state-of-the-art flood modelling technology supported by parallelised high-performance computing, and object-based GLOF exposure and impact evaluation using open-source data. Not covered in Veh & Walz (2020), our study conducts high-resolution inundation simulations for various GLOF scenarios using flood modelling technology. This facilitates subsequent object-based assessments of GLOF exposures and impacts. The results provide a comprehensive and detailed evaluation of potential exposures and impacts stemming from these PDGLs. While much of the prior work has focused on the initial step of GLOF risk evaluation, specifically hazard assessment for glacial lakes, like Veh & Walz (2020), our study advances the field by addressing the second stage, which involves exposure and impact evaluation. The insights gained from this study can empower authorities not only with knowledge of where threats exist but also with an understanding of the expected magnitude and location of impacts.

I agree that incorporating additional data, such as moraine dam geometry (height and width), would enhance the estimation of maximum breach depth and flow volume. However, it is challenging to obtain these data for all 21 lakes. Therefore, to encompass all potential glacial lake outburst scenarios, we have also considered less severe conditions. Specifically, scenarios where 25%, 50%, and 75% of the lake water volume is released have been examined. The outcomes of these less severe scenarios have been compared to the worst-case conditions, where 100% of the lake water is released, as discussed in Section 4.3.1.

Further, the procedure of random selection of 1000 scenarios and subsequent calculation of inundation frequencies and median of max. inundation depths for each lake is not appropriate

because these scenarios are not equally probable. Reflecting on frequency-magnitude relationships of common GLOF triggers (various mass movements), low to moderate magnitude GLOFs are more frequent and more likely while extreme GLOFs are rare and less likely. Instead of selecting the scenarios randomly, my suggestion is to select them on purpose to cover the full range, with assigned weights (or ideally probabilities).

Response: Thank you so much for pointing out this question. When taking the median of the maximum values, we default to assuming each scenario is equally probable. However, as highlighted in the reviewer's comments, this assumption is not correct. Therefore, we assigned weights to each scenario based on probabilities. The weight of each scenario is determined by its occurrence probability, i.e., the proportion of times its peak discharge does not exceed that of other scenarios relative to the total number of scenarios. A smaller proportion indicates a lower likelihood of occurrence, while a larger proportion indicates a higher likelihood. The weight of each scenario is calculated by dividing the proportion by the total proportion of all possible scenarios. Subsequently, the final flood inundation probability and maximum water depth are derived by multiplying each scenario's results by its respective weight. Based on these derived flood inundation probability and maximum water depth values, exposure and impact evaluations have been conducted for these 21 lakes. Section 4.3, which covers flood inundation simulation, exposure, and damage assessment, has been reanalysed and rewritten.

Since the modelling part lacks any validation, this is where frequency-magnitude relationship can come into play. I wonder whether employing your approach over past GLOFs can yield "typical extremity" of GLOFs (if you standardize the extremity of your scenarios for each lake on the dimensionless scale from 0-1)? While it is mentioned in Discussion section that the incompleteness of data about past GLOFs prevents the authors from attempting validation, I wonder whether any single GLOF characteristic (e.g., breach depth, flood volume, peak discharge, inundation area, etc.) could be used to validate the flood modeling results and estimate "typical extremity" of GLOFs in Nepal? Such an analysis could guide the weighting of your scenarios.

Response: On one aspect, we did not consider the GLOF outburst frequency because the underlying database for frequency–magnitude relations typically is very poor. Here, we considered different scenarios, i.e., the release of lake water at different predetermined proportions. Under the predetermined proportion, we examined a plausible range of values for lake volumes and peak discharges for each glacial lake, ensuring comprehensive coverage of all potential glacial lake outburst scenarios.

For validation, we appreciate the reviewer's understanding regarding the lack of historical event records, which is a common issue with GLOF inundation simulations. Additionally, it is noteworthy that our proposed framework utilises the fully physically based hydrodynamic model HiPIMS, intricately designed to capture the highly transient and complex hydrodynamic processes induced by events such as dam breaks and flash floods. HiPIMS has been successfully validated for these extreme flow conditions. The adoption of this model enhances our confidence in simulating the spatial-temporal processes of GLOF inundation, ultimately contributing to improved hazard evaluation results. Regarding the "typical extremity" of GLOFs, the weight assigned to each scenario is determined by its occurrence probability (see above). We are uncertain whether this addresses the reviewer's comments, and we would greatly appreciate further guidance and clarification on the term "typical extremity".

Overall, I'm in favor of recommending this study for further processing and subsequent publication after some modifications are considered. My suggestions to the authors are: (i) to

consider dam geometry when estimating max. flood volume; (ii) to consider the validation of this approach with some of the past GLOFs in the country (and obtaining "typical extremity"); (iii) to consider avoiding the use of random selection of scenarios which may be misleading.

Response: Thank you very much for the valuable comments from the reviewer. If there are any additional considerations beyond our responses provided above, we hope the reviewer will let us know.

---

## Referee Report (RR1)

The authors reflected on my comment regarding dam geometry and introduced a characteristic named "dam depth" ("dam height" may be more suitable). However, it is not appropriate to consider that dam height is identical to maximum breach depth. The worst moraine dam breaches reported in the literature are up to several tens of meters deep, however, the authors consider > 100 m for most of the lakes, resulting in the overestimation of what is called the worst case scenario. Here I use the the example of Lower Barun lake to illustrate why:

The authors estimated the dam height of this lake 128 m (from the toe to the crest) and consider this value the maximum breach depth. Now let's have a look at the lake bathymetry from Gantayat et al. (2024; https://doi.org/10.1016/j.scitotenv.2024.175028):

[Figure]

I draw 128 m deep breach as dashed red line:

[Figure]

Such a breach (here I draw a profile) would have to be > 2,500 m long while its slope would have to be 0° at the same time (dashed red line):

[Figure]

Both these assumptions are unrealistic. In reality, the inclination of the breach channel will be > 0° (schematically represented by dashed green line). And this makes a huge difference. Further, since the lake occupies overdeepened depression, it is unlikely that the dam is even erodible to such depth (bedrock is likely at certain depth or the erosion would stop because of too low slope inclination of the channel).

Therefore, I argue that max. breach depths, flood volumes as well as peak discharges of all lakes with assumed breach depths > 100 m (and resulting flood extents downstream) are overestimated rather than "worst case".

I'm convinced that this issue should be addressed (or at least acknowledged) before the study is accepted for publication. Thank you.

Adam Emmer
University of Graz, Austria

---

## Author Response (AR2)

** Please find our point-by-point response highlighted in blue to the reviewer's comments.

**Response to Referee #1: Adam Emmer**

I thank the authors for the revisions of their manuscript and their detailed replies to my comments. While I understand why some of my comments were not reflected in the revised version, it is in particular disappointing to see that the authors refused to take into account dam geometry which controls breach depth and so released volume, and which can be obtained easily from digital elevation models. The assumption of up to 100% water release is not justified if the dam geometry is not considered and I'm very reluctant to arbitrarily calling it "the worst-case scenario". It does not justify it. An effort should be made to set the boundary conditions of a "worst case scenario" (otherwise you can model collapses of whole mountains, calling it WCS; such outcomes, however, have little utilisation as the probability of their occurrence in human-relevant timeframes is unknown). And considering the dam geometry is one of the ways how a boundary conditions for the DRR-relevant worst case scenario definition can be set up. And this also aligns with my comments about "typical extremity" which can be expressed - for instance - as % of total lake volume that is released during the GLOF. My experience is that GLOFs from moraine-dammed lakes typically do not involve 100% of lake water, suggesting boundary conditions for a WCS. To sum up, I ask the authors to justify their assumption that a WCS for any lake means a release of the 100% of the lake water (I'm not sure there is a data-driven evidence and justification for that), or consideration of dam geometry.

Response: Thank you very much for the reviewer's comments. Although considering dam geometry to set up the worst-case scenarios would require re-simulating all scenarios, which involves substantial work, we fully agree with the reviewer's suggestion. We have contacted the authors (Bajracharya et al., 2020), who assessed GLOF hazard factors related to dam characteristics for the 21 identified PDGLs and have obtained the maximum depth of the moraine dams for these lakes. To account for the most severe GLOF scenarios, we have considered the breach depth to reach the maximum depth of the moraine dams. Additionally, we have also accounted for less severe conditions by simulating scenarios where 25%, 50%, and 75% of the maximum dam depth are breached. The main revisions can be found in the Methodology subsection 2.2.1 (lines 196–219), the data in Table 1 (lines 306–307), and the Results subsections 4.2 and 4.3.

**Response to Referee #2**

Dear authors,

Thanks for the detailed responses to my comments.

I read both original and the revised version. The authors have addressed my main concerns and made great improvements on the initial submission. The paper is more clearly written and more concise. But I still have three concerns for the paper although much more minor than the first submission.

Response: Thank you very much for the reviewer's overall positive feedback.

First, the authors have deleted the introduction of the three categories of glacial lakes in the text but the classification still remains in Fig. 2. I suggest to remove the classes of the lakes in this figure. Instead, the authors can mark the lake numbers in Fig. 2, if possible. The risk classification of the lakes is also seen somewhere in the text, e.g., in section 4.3.3. The authors need to go through the paper to remove the classification in the text.

Response: Figure 2, shown below, has been improved based on the reviewer's comments. The risk classification of the lakes has been reviewed and removed from the draft.

[Figure]

Fig 2. Study area and 21 identified dangerous glacial lakes each with a unique lake number, and potentially impacted hydropower plants.

The second issue is related to the hydropower plants. It is unclear how the hazard and damage to the hydropower stations was evaluated. It seems that the authors only considered whether the GLOFs would inundate the hydropower plants downstream. I believe the reality is more complicated. Inundation, flow impact, and sediment deposition may all cause damage to the hydropower stations. The influence of GLOF on the operation of the plants can also lead to economic loss although the dam and plant remain intact in the flood. The damage curves were utilized for buildings and roads but it seems that no damage curve was there for the hydropower stations. Whether and how the damage of the hydropower engineering was considered needs to be clarified.

Response: Thank you to the reviewer for highlighting this important point that we did not address. We have added clarification below in lines 503–513.

"In this study, we have identified 49 existing and planned hydropower projects that could potentially be impacted by GLOFs originating from the 21 PDGLs; however, we did not assess the specific impacts of GLOFs on these hydropower projects. To our knowledge, there are no readily available damage curves that correlate the potential impact on hydropower plants with flood depth and other flood characteristics. Furthermore, hydropower plants typically comprise multiple components, including the dam and reservoir, powerhouse and auxiliary facilities, among others. The spatial extent of a hydropower plant can vary significantly, ranging from a few square kilometres to several hundred square kilometres. Accurate assessment would require detailed spatial information and mapping of hydropower plants, which is currently lacking. Consequently, this study focuses exclusively on identifying whether a hydropower plant is potentially at risk from GLOFs, without engaging in a detailed assessment of the specific damages that may be incurred."

Besides, the hydropower stations listed in Table 4 can also be shown in Fig. 2 to visualize the exposure of these hydropower stations to GLOFs. Further information is better to be shown in Table 4, e.g., capacity, commission/issue date, corrected longitude and latitude, how impacted by nearby glacier lake(s). Lines 425-429 should be moved to the methods section, combined to the paragraph from line 256 to line 272. The paragraph in lines 425-429 also needs to be shortened. For instance, the sentence in lines 430-433 can be rewritten if the information is shown in Table 4 somehow.

Response: These hydropower stations have been added to Figure 2, and detailed information such as capacity, commission/issue date, corrected longitude and latitude, and the lake(s) that may pose risks have been included in Table S1 of the supporting document. The original lines 425–429 have been moved to lines 270–272. The paragraph in lines 425–429 has been shortened as follows.

"In addition to the high potential for human settlements to be exposed to GLOFs, hydropower projects are increasingly vulnerable to these events. A total of 49 hydropower plants have been identified as being in close proximity to GLOF flow channels, thereby rendering them potentially vulnerable to GLOFs associated with the 21 PDGLs. Among these, 5 plants are currently operational. Additionally, 44 hydropower plants, for which generation or survey licenses have been issued, are also exposed to the risk of GLOFs from these 21 PDGLs. When examining the potential impact of lakes on operational hydropower plants and those holding generation licenses, it is observed that Thulagi and Tsho Rolpa pose a risk of inundating 5 plants (3 operational and 2 licensed) and 3 plants (all licensed), respectively. Moreover, it is noteworthy that lakes Anonymous 12, Anonymous 1, and Anonymous 2 have the potential to inundate 7 plants (2 operational and 6 licensed), 2 plants (both licensed), and 2 plants (both licensed), respectively."

Last, the authors showed the statistics on the damage to buildings, roads and agriculture land caused by GLOFs, but did not visualize the spatial distribution of this damage in the areas subjected to GLOFs. Maybe this has been shown in previous work of the authors, but providing some examples of the damage maps will make the advantages of the advanced hydrodynamic simulating technique clearer to the audience. The authors can consider to provide such examples in appendices or supplementary material if putting them in the main text is inappropriate.

Response: An example figure has been added. Figure 8 uses Lake Anonymous 12 to illustrate the spatial distribution of damage to buildings, roads, and agricultural land caused by GLOFs.

[Figure]

Fig 8 Damage to buildings, roads, and agricultural land caused by the most serious GLOF due to Lake Anonymous 12

---

## Author Response (AR3)

\*\* Please find our point-by-point response highlighted in blue to the reviewer's comments.

**Response to Referee #1: Adam Emmer**

The authors reflected on my comment regarding dam geometry and introduced a characteristic named "dam depth" ("dam height" may be more suitable). However, it is not appropriate to consider that dam height is identical to maximum breach depth. The worst moraine dam breaches reported in the literature are up to several tens of meters deep, however, the authors consider > 100 m for most of the lakes, resulting in the overestimation of what is called the worst case scenario. Here I use the example of Lower Barun lake to illustrate why: The authors estimated the dam height of this lake 128 m (from the toe to the crest) and consider this value the maximum breach depth. Now let's have a look at the lake bathymetry from Gantayat et al. (2024; https://doi.org/10.1016/j.scitotenv.2024.175028):

[Figure]

I draw 128 m deep breach as dashed red line:

[Figure]

Such a breach (here I draw a profile) would have to be > 2,500 m long while its slope would have to be 0° at the same time (dashed red line):

[Figure]

Both these assumptions are unrealistic. In reality, the inclination of the breach channel will be > 0° (schematically represented by dashed green line). And this makes a huge difference. Further, since the lake occupies overdeepened depression, it is unlikely that the dam is even erodible to such depth (bedrock is likely at certain depth or the erosion would stop because of too low slope inclination of the channel).

Therefore, I argue that max. breach depths, flood volumes as well as peak discharges of all lakes with assumed breach depths > 100 m (and resulting flood extents downstream) are overestimated rather than "worst case".

I'm convinced that this issue should be addressed (or at least acknowledged) before the study is accepted for publication. Thank you.

Response: We sincerely appreciate Dr Emmer's valuable feedback. We agree with Dr Emmer's perspective that, for certain lakes, a complete dam breach may be unrealistic due to factors such as the inclination of the breach channel and the specific lithology, composition, and structural characteristics of the dam. Accurately depicting the most severe plausible scenario for each lake requires detailed dam information, and in situ investigations remain the most reliable approach to estimate potential breach sizes. However, obtaining such precise information for all 21 lakes may exceed the scope of this study. Accordingly, we acknowledge this limitation in the manuscript and have added discussion in lines 458-470, including the clarification: "It is recognized that for certain lakes, a complete (100%) breach may be improbable and represents only a theoretical worst-case scenario. In practical terms, the most severe realistic scenario should consider the unique lithology, composition, and structural characteristics of each moraine dam". Additionally, we have reviewed the manuscript to ensure consistency, replacing terms such as "the worst situation" with "theoretical worst situation/scenarios, i.e., a complete breach of dam height". Beyond the 100% breach scenario, we evaluated breaches at 10%, 30%, and 50% of dam height in section 4.3.1, aiming to provide a comparative perspective for each lake and offer more practical insights. 'Dam depth' has been revised to 'dam height' throughout the manuscript.

**Lines 458 – 470**: We evaluate GLOF scenarios involving breaches of 10%, 30%, 50%, and 100% of dam heights. It is recognized that for certain lakes, a complete (100%) breach may be improbable and represents only a theoretical worst-case scenario. In practical terms, the most severe realistic scenario should consider the unique lithology, composition, and structural characteristics of each moraine dam; however, conducting such detailed filed investigation to gather this information across multiple lakes at a large scale remains challenging. For large-scale GLOF risk assessments, Zhang et al. (2023) applied an empirical relationship between lake volume and flood volume, derived from historical GLOFs, to estimate flood volumes, capping the maximum flood volume at $20 \times 10^6 \, \text{m}^3$ due to limited data on large glacial lakes. Fujita et al. (2013) estimated potential flood volume by analysing the depression angle from lake shorelines using DEM data, noting that potential flood volume is helpful for preliminarily identifying and prioritising lakes for further investigation but does not directly quantify GLOF risk. As no straightforward and reliable method currently exists for accurately predicting flood volumes across multiple lakes, we analysed scenarios assuming breaches at 10%, 30%, 50%, and 100% of dam heights for consistency. When interpreting these impact results, the inherent limitations in predicting flood volume and the realistic likelihood of each scenario should be carefully considered.

**Response to Referee #2**

Dear authors, Thanks for carefully responding and addressing my comments. The paper was further improved in last revision and I don't have further concerns except for some language/technical issues. I put some examples below and suggest the authors to go through the paper again.

Response: Thank you to the reviewer for the continued support in helping us improve the manuscript. We have thoroughly reviewed the paper to enhance clarity and fluency, remove unnecessary information, and correct inaccuracies in expression.

Line 16: is it necessary to keep 'derived from previous research'?

Response: We have removed 'derived from previous research'.

Lines 26-28: I suggest to rewrite the last two sentences to better summarize the results. Then the coordinates can be removed from the abstract.

Response: Six anonymous lakes (located at 85°37'51" E, 28°09'44" N; 87°44'59" E, 27°48'57" N; 86°55'41" E, 27°51'00" N; 86°51'29" E, 27°41'13" N; 86°55'01" E, 27°49'55" N; 87°56'05" N, 27°47'26" E) have the potential to impact more than 200 buildings. Moreover, anonymous lake (located at 85°37'51" E, 28°09'44" N) have the potential to inundate existing hydropower facilities. → Rewrite to 'One anonymous lake in the Trishuli River Basin, two anonymous lakes in the Tamor River Basin, and three anonymous lakes in the Dudh River Basin have the potential to impact more than 200 buildings. Moreover, the anonymous lake in the Trishuli River Basin has the potential to inundate existing hydropower facilities.'

Line 56: change 'has' to 'have'.

Response: It has been changed.

Fig. 3: I don't see the need to keep panel (a) as: (i) it still contains lake classification, and (ii) the lake indices have been shown in fig. 2.

Response: We have removed panel (a) and now display six panels below in Fig. 2.

[Figure]

Fig. 8: Scales are needed for all the panels.

Response: Scales has been added in Fig. 8.

[Figure]

Table S1: what is the unit for capacity?
Response: The unit is megawatt (MW), which has been added to Table S1.

---

## Author Response (AR4)

**Response to Editor: Damien Bouffard**

Dear authors, thanks for the revised version. I have one last minor point: I suggest to change "anonymous lake" by "unnamed lake" in abstract, text and figures

Response: The changes have been made in abstract, text and figures.